# Phenotypic and genetic characteristics of retinal vascular parameters and their association with diseases

Sofía Ortín Vela [1,2,11] ✉, Michael J. Beyeler [1,2,11] ✉, Olga Trofimova[1,2], Ilaria Iuliani[1,2], Jose D. Vargas Quiros[3,4], Victor A. de Vries [3,4], Ilenia Meloni[5,6], Adham Elwakil[5,6], Florence Hoogewoud[5], Bart Liefers[3,4], David Presby[1,2], Wishal D. Ramdas [3], Mattia Tomasoni [5,6], Reinier Schlingemann[5,7], Caroline C. W. Klaver [3,4,8,9] & Sven Bergmann [1,2,10] ✉

Fundus images allow for non-invasive assessment of the retinal vasculature whose features provide important information on health. Using a fully automated image processing pipeline, we extract 17 different morphological vascular phenotypes, including median vessels diameter, diameter variability, main temporal angles, vascular density, central retinal equivalents, the number of bifurcations, and tortuosity, from over 130,000 fundus images of close to 72,000 UK Biobank subjects. We perform genome-wide association studies of these phenotypes. From this, we estimate their heritabilities, ranging between 5 and 25%, and genetic cross-phenotype correlations, which mostly mirror the corresponding phenotypic correlations, but tend to be slightly larger. Projecting our genetic association signals onto genes and pathways reveals remarkably low overlap suggesting largely decoupled mechanisms modulating the different phenotypes. We find that diameter variability, especially for the veins, associates with diseases including heart attack, pulmonary embolism, and age of death. Mendelian Randomization analysis suggests a causal influence of blood pressure and body mass index on retinal vessel morphology, among other results. We validate key findings in two independent smaller cohorts. Our analyses provide evidence that large-scale analysis of image-derived vascular phenotypes has sufficient power for obtaining functional and causal insights into the processes modulating the retinal vasculature.

The retina provides a unique opportunity for imaging human vasculature. In particular, retinal colour fundus images (CFIs) allow for noninvasive in-vivo assessment of the vascular system of the superficial inner layer of the retina. Such images have been acquired in several cohorts and there is a large body of research on automatic extraction of vascular properties and their associations with medically relevant information.

It is well established that vascular properties obtained from retinal imaging not only enable the monitoring of ocular diseases such as diabetic retinopathy, macular degeneration, and glaucoma, but can also serve as a powerful screening tool for early detection of systemic diseases, including stroke[1-4], coronary heart disease[5,6], peripheral artery diseases[7], hypertension[3,4,8-18], atherosclerosis[2,3,19], and myocardial infarction[20,21]. Retinal abnormalities have also been

associated with common comorbidities such as diabetes[8,22–24] and obesity[25].

The processing of CFIs typically can be divided into three steps. First, the image is processed at the level of pixels to identify which of them represents blood vessels (possibly distinguishing between arteries and veins[26–28]), or other structures, like the optic disc (OD). In the second step this pixel-wise information is used to annotate the retina in terms of objects, such as vessel segments represented as a list of points along their midline, as well as the vessel widths along these points. Finally, the information from these objects is used to measure different vascular properties such as vessel diameter or tortuosity, number of bifurcations, as well as certain angles between major vessels. Some simple vascular phenotypes, such as vascular density or fractal dimension, can also be computed directly from the pixel-wise information.

While there have been several studies analysing retinal vascular phenotypes[29–34], most of them focused on measuring just one or few retinal phenotypes, often in small image sets, and some required expert input rather than being fully automated[35–37]. Furthermore, the software used for vascular phenotyping is usually not openly accessible, with two recent exceptions[38,39]. Together this precludes the establishment of a comprehensive and reproducible characterisation of large retinal image collections.

Recently, Deep Learning (DL) approaches have gained popularity in retinal image analysis. The first contribution is at the level of pixel-wise annotation, where state-of-the-art segmentation can be achieved with Convolutional Neural Network (CNN) architectures. For example, the little W-Net (LWNET) annotates pixels as being part of an artery or vein, outperforming classical segmentation approaches[40]. Such networks can also be used to annotate pixels belonging to the OD[41], vessel bifurcations[42], or other structures. The second contribution of deep CNNs is to learn latent variables providing efficient low-dimensional representations of retinal images[43,44]. Such self-supervised, image-based phenotyping can generate additional phenotypes complementing explicit retinal features. Finally, DL approaches have been used to directly predict health-relevant phenotypes from retinal images[45–47].

Beyond their value for assessing ocular or systemic health, retinal phenotypes have also been used in Genome-Wide Association Studies (GWAS) to identify genetic variants modulating these phenotypes. However, previous GWAS have focused on a limited set of vascular properties, such as vessel diameter[28,48–50], tortuosity[26,27], vascular density, fractal dimension[51], and certain deep latent variables[43] (see Supplementary Discussion "Comparison with previous GWAS", and Supplementary Table 1).

In this study, we present a joint analysis of 17 retinal vascular phenotypes, including features like temporal angles, number of bifurcations, or diameter variability, which have not been analysed previously at a large scale. Leveraging data from the UK Biobank (UKBB), and employing our open-source fully automated analysis platform, we provide retinal vascular phenotyping for close to 72 k subjects, after quality control (QC). For all phenotypes, we performed GWAS, heritability estimates, gene and pathway analyses, as well as associations with a broad set of systemic and ocular diseases. This allowed us to compare cross-phenotype correlations, both at the phenotypic and genotypic levels, study in detail which genes affect individual or multiple phenotypes, and identify potential causal relationships with diseases (see Fig. 1 depicting the overall methodology of our discovery study). We reproduced the phenotypic correlation structure in two independent smaller cohorts: the Rotterdam Study (RS, $N = 8.1$k) and OphtalmoLaus ($N = 2.2$k)[52,53]. Analysis of RS data also provided consistent estimates for phenotypes heritabilities and genetic correlations, as well as replication of a large number of our genome-wide significant hits for the vast majority of phenotypes.

## Results

### Automated pipeline for phenotyping of the retinal vasculature

To analyse the retinal vasculature, we developed an automated pipeline building on our previous work on retinal vessel tortuosity[27]. This pipeline enabled us to segment and annotate the retinal vasculature and OD from 130 361 colour fundus images (CFIs) of 71 494 subjects after Quality Control (QC). We used a previously established method for computing a QC score[51], and removed images in the lowest quartile of this score. Applying a range of QC thresholds, we observed that phenotype heritabilities tended to be higher, while disease incident rates were lower when applying more stringent thresholds, possibly because some diseases impact the vascular density which is a good proxy for image quality (see Supplementary Figs. 1–3 "UKBB Quality control threshold effect on results" for details).

We selected 17 representative image-derived phenotypes (IDPs) from a broader set of 36 phenotypes to characterize each image, including vascular densities, median tortuosities, central retinal equivalents, median diameters, diameter variabilities, and main temporal angles for arteries and veins. We also calculated the ratios between artery and vein values for the first four phenotypes, and we estimated the total number of vessel bifurcations. These IDPs were chosen based on their associations with diseases and the reliability of measurement. The distributions of these main IDPs are presented in Supplementary Fig. 4 "UKBB Distribution of retinal vascular IDPs". The broader set of 36 IDPs can be found in Supplementary Methods "Phenotype extraction".

### Correlation structure and heritabilities of retinal vascular IDPs

To explore the relationships between our main IDPs, we calculated pair-wise phenotypic Pearson correlations, $r^{(p)}$ (upper right triangle in Fig. 2a). We observed that the same IDPs measured for arteries and veins tended to cluster together, in particular for the temporal angles, tortuosities, and vascular densities. Vascular densities were highly correlated with each other and the number of bifurcations, the strongest correlation observed. They also correlated with venous, but not arterial, tortuosity and anti-correlated with median diameters. The temporal angles exhibited low correlations to the other phenotypes. Finally, most artery-vein ratios were highly correlated with their corresponding arterial, and highly anti-correlated with their corresponding venous measures.

To assess the genetic relationships between the IDPs, we estimated pair-wise genetic correlations, $r^{(g)}$, using cross-trait Linkage Disequilibrium Score Regression (LDSR)[54] (lower left triangle in Fig. 2a). On average, genetic correlations were slightly higher than phenotypic correlations (standardized mean difference across the 136 pairs ($ij$) is $d = 0.34$, t(135) = 6.53, $p = 6.2 \times 10^{-10}$). The largest difference occurred between the temporal angles. Overall there was a strong correspondence between genetic and phenotypic correlations, $corr(r^{(g)}, r^{(p)}) = 0.86$, permutation $p < 1 \times 10^{-4}$. Please refer to "Correlation structure and heritabilities of extended list of retinal vascular IDPs" in Supplementary Methods for the corresponding analysis for our broad set of 36 IDPs. The exact values of genetic and phenotypic correlations are available on Figshare.

To gain deeper insights into the genetic control of retinal vascular morphology, we estimated the single nucleotide polymorphism (SNP) heritability, $h^2$, for each IDP (Fig. 2b), using LDSR[55]. Our analysis revealed varying levels of heritability across our IDPs. Arterial tortuosity exhibited the highest heritability, $h^2 = 0.25 \pm 0.02$, followed by the tortuosity ratio, $h^2 = 0.20 \pm 0.02$, and venous diameter variability, $h^2 = 0.18 \pm 0.02$. In contrast, the median diameter measures, particularly arterial median diameter, showed the lowest heritability, $h^2 = 0.05 \pm 0.01$ (see Supplementary Fig. 5 and 6 for Manhattan and Quantile-Quantile plots from the GWAS of each IDP).

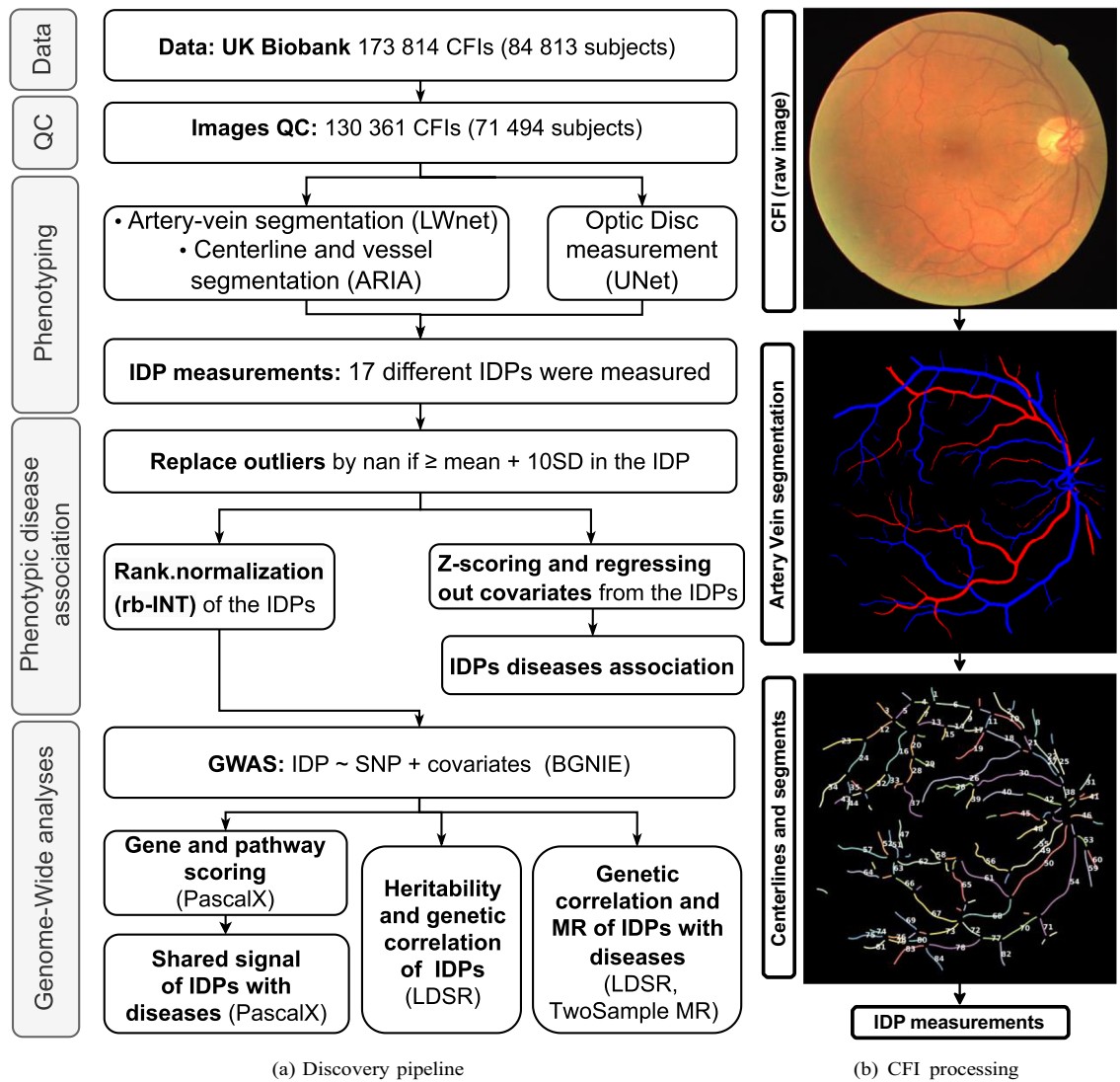

(a) Discovery pipeline

(b) CFI processing

**Fig. 1 | Discovery pipeline and measurement of retinal vascular image-derived phenotypes (IDPs). a** Overview of discovery pipeline. Subjects' basic and medical information, genotypes, and CFIs were collected from the UKBB. Applying the Image QC method of[51] removed ~25% of all CFIs. Pixel-wise vessel segmentation and classification were performed using LWNET[40]. ARIA[77] was used to identify vessel segment objects. A DL network was used to measure the position of the OD[41]. Based on this primary information our bespoke algorithms measured vascular IDPs. IDPs (z-scored and corrected for covariates) were associated with diseases through linear and logistic regressions, and Cox models. GWAS was performed on all IDPs after rank-based inverse normal transformation (rb-INT) and correction for covariates, and the resulting summary statistics were used to estimate heritabilities and genetic correlations, to identify relevant genes and pathways, and to study the genetic association and potential causal relationships between the IDPs and some of the disease phenotypes. **b** Overview of IDP measuring process. Top: original CFI from DRIVE dataset. Middle: Pixel-wise segmented vasculature with artery-vein classification using LWNET[40]. Bottom: Vessel segment objects in terms of centrelines and diameters were identified using ARIA[77], providing the starting point for measuring vascular IDPs (see Supplementary Methods "Vesssel segmentation", "Optic disc segmentation", and "Phenotype extraction" for details).

## Gene and pathway level analysis of retinal vascular IDPs

To identify the genes modulating specific vascular IDPs, we employed our *PascalX* analysis tool[56,57]. This tool aggregates SNP-wise association signals within gene windows and generates gene scores.

The number of genes associated with IDPs (diagonal of Fig. 3a) ranged from 1 gene for the arterial median diameter to 252 genes for arterial tortuosity. We observed the highest number of genes for tortuosities, venous diameter variability, and venous central retinal equivalent. The off-diagonal elements in Fig. 3a show the number of common genes for each pair of IDPs. Generally, more correlated IDPs tended to share more genes. However, even among highly similar IDPs, a significant portion of genes were phenotype-specific, and some IDPs shared only a few or no genes.

While no single gene was shared among all IDPs, two genes, *LINC00461* and *CTC-498M16.4*, were associated with 9 of the 17 IDPs

(Fig. 3b). Other genes associated with multiple IDPs included *SIX6*, *FLT1*, *FUT1*, *HERC2*, and *PDE6G*. The complete list of significant genes associated with each IDP is available on Figshare.

Furthermore, we conducted pathway analysis using *PascalX* to identify gene sets, or "pathways", that exhibited higher association signals for each IDP GWAS than expected by chance. Although no single pathway was shared among all IDPs, some of the most frequent pathways included 'Fetal retina fibroblast', and 'Abnormal retinal morphology' (Supplementary Note 1). The complete list of significant pathways per IDP is available on Figshare.

To further investigate pleiotropic genes, we employed *PascalX* cross-GWAS analysis[57], a method that examines coherent effects across the SNPs associated with two phenotypes within a gene window. This approach has more power than just intersecting the gene sets of two individual phenotypes and also allows for distinguishing between

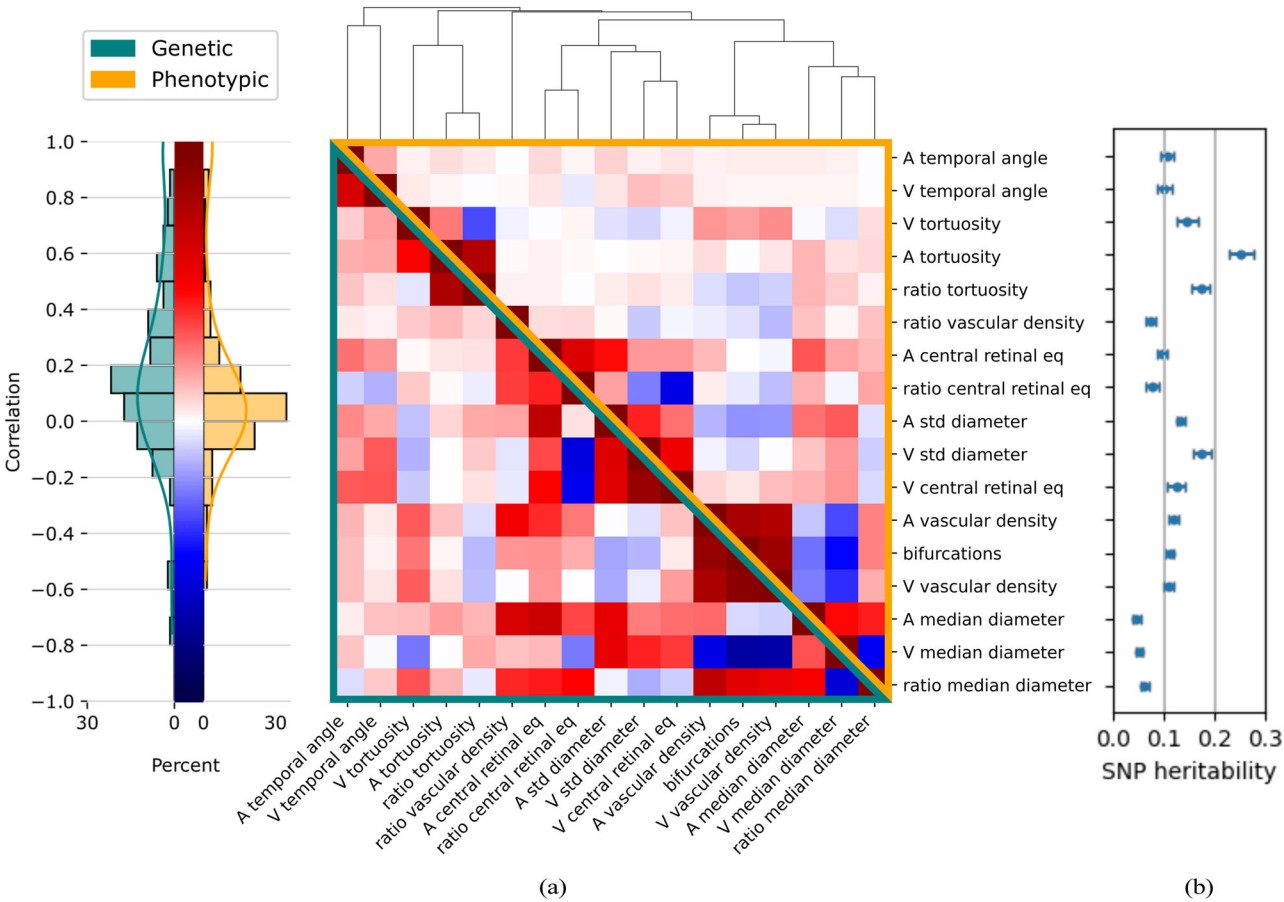

(a)                                                                                          (b)

**Fig. 2 | Phenotypic and genetic correlations of retinal vascular IDPs and their heritabilities. a** Phenotypic (upper-right orange triangle) and genetic (lower-left green triangle) correlations between retinal vascular phenotypes, clustered by absolute phenotypic correlation distance, $1 - |corr|$ . The 17 phenotypes are (A: artery, V: vein): main temporal angles ('A/V temporal angle'), median tortuosities and their ratio ('A/V tortuosity' and 'ratio tortuosity'), central retinal equivalents and their ratio ('A/V central retinal eq' and 'ratio central retinal eq'), diameter variabilities ('A/V std diameter'), vascular densities and their ratio ('A/V vascular density' and 'ratio vascular density'), median diameters and their ratio ('A/V median diameter' and 'ratio median diameter'), and the number of bifurcations ('bifurcations'). Phenotypes were corrected for age, sex, eye geometry, batch effects, and ethnicity before phenotypic clustering and before GWAS (see Methods). **b** Corresponding phenotype SNP

heritabilities, $h^2$, and their standard error, estimated using LDSR[55]. In LDSR, heritabilities are estimated as the OLS slope from regressing the mean Chi-squared statistics of SNPs onto their corresponding LD scores, while accounting for the sample size and the total number of SNPs. Sample sizes (phenotypic and genetic respectively): A temporal angle [55.3 k, 54.9 k], V temporal angle [57.9 k, 57.5 k], A tortuosity [68.6 k, 68.1 k], V tortuosity [68.5 k, 68.0 k], Ratio tortuosity [68.5 k, 68.0 k], A central retinal eq [65.5 k, 65.0 k], V central retinal eq [65.8 k, 65.4 k], Ratio central retinal eq [64.9 k, 64.4 k], A std diameter [68.5 k, 68.0 k], V std diameter [68.5 k, 68.0 k], Bifurcations [68.2 k, 67.8 k], A vascular density [68.7 k, 68.2 k], V vascular density [68.7 k, 68.2 k], Ratio vascular density [68.4 k, 68.0 k], A median diameter [68.6 k, 68.1 k], V median diameter [68.5 k, 68.0 k], Ratio median diameter [68.5 k, 68.0 k].

coherent and anti-coherent effects. However, even in this more sensitive analysis, no single gene was shared between all pairs of phenotypes. The most pleiotropic genes largely overlapped with those identified by simple gene-scoring, yet they tended to be shared among more IDPs (Fig. 3c, d). The complete list of PascalX IDP-IDP cross gene scores is available on Figshare. Generally, IDP pairs with positive LDSR genetic correlation shared more coherent gene signals, while those with negative LDSR genetic correlation had predominantly anti-coherent signals (see Supplementary Fig. 7 "UKBB LDSR genetic correlation against PascalX").

**Phenotypic association with diseases and risk factors**
To evaluate the clinical relevance of retinal vascular morphology, we examined the phenotypic associations between the retinal vascular IDPs and various eye-related diseases, vascular diseases, and their associated risk factors. We employed linear regression for continuous diseases variables (that correspond to risk factors) (Fig. 4a), logistic regression for binary disease states (Fig. 4b), and Cox models for

diagnoses with age-of-onset (Fig. 4c). All IDPs were standardized (z-scored) and adjusted for potential confounders.

Amongst the risk factors, diastolic blood pressure (DBP) and systolic blood pressure (SBP) displayed similar associations with most IDPs. Smoking pack-years exhibited the strongest positive association with venous diameter variability, $\beta = 0.12$, while the strongest negative correlations were observed between blood pressure (BP) and arterial-related IDPs, particularly the central retinal equivalent, vascular density, and median diameter $\beta \in [-0.18, -0.13]$ (Fig. 4a).

For binary disease states, hypertension was positively associated with venous tortuosity and less with arterial tortuosity, while negatively associated with central retinal artery equivalent, and the ratios of central retinal equivalents and vascular densities, among others. Eye diseases such as amblyopia were negatively associated with the number of bifurcations and venous vascular density, $\beta \approx -0.16$. Others like presbyopia, hypermetropia, and myopia showed smaller effects, $\beta \in [-0.08, 0.06]$, but were still statistically significant. Retinal diabetes was negatively associated with the vascular densities and the number

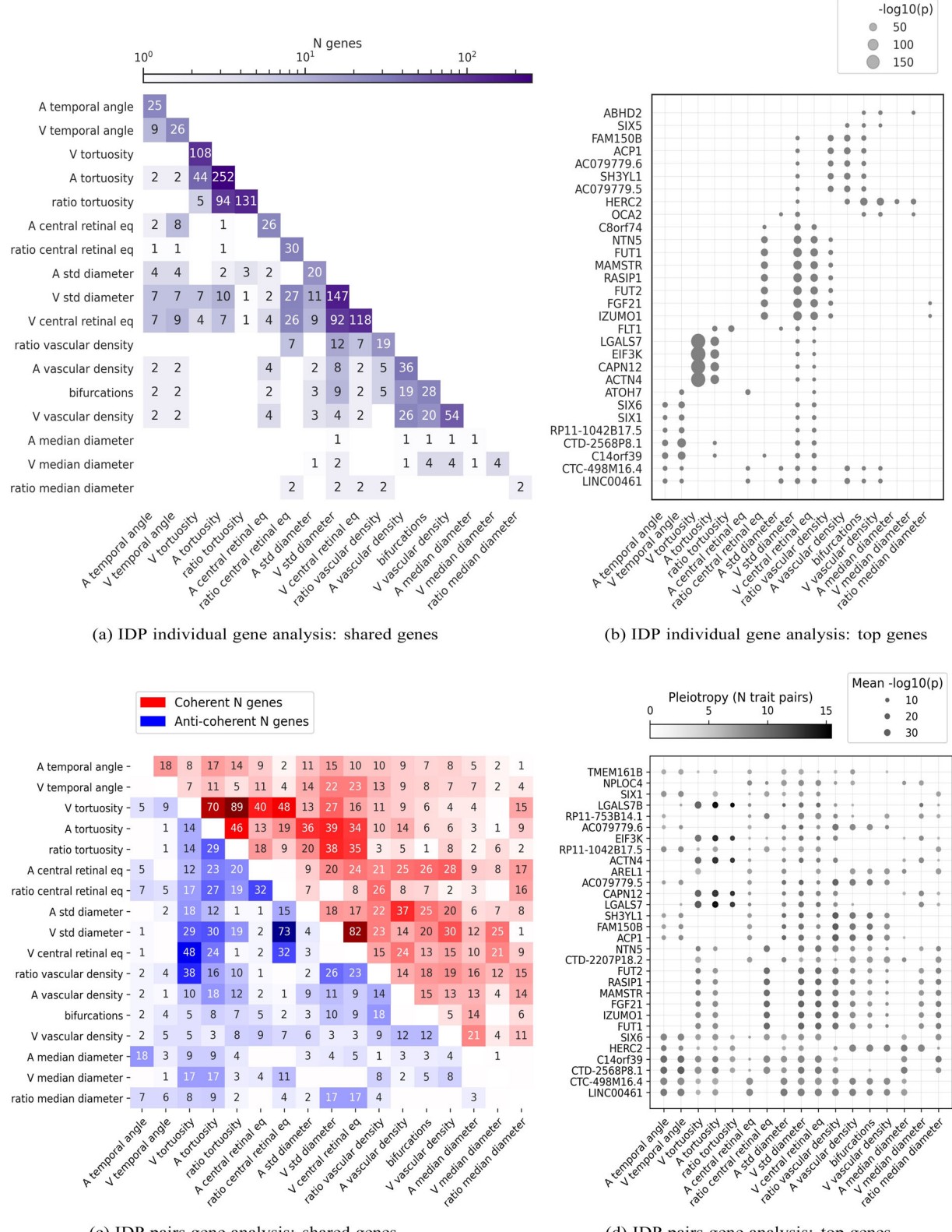

(a) IDP individual gene analysis: shared genes

(b) IDP individual gene analysis: top genes

(c) IDP pairs gene analysis: shared genes

(d) IDP pairs gene analysis: top genes

**Fig. 3 | Gene associations with vascular IDPs. a** Number of genes associated with the different IDPs. The diagonal shows the number of genes significantly associated with each IDP. The lower triangle shows the number of genes in the intersection between pairs of IDPs. **b** 30 genes most frequently associated with the IDPs. Dot sizes are inversely proportional to *p*-values. **c** Number of genes showing coherent (top right) or anti-coherent (bottom left) signal between pairs of IDPs. **d** 30 genes most frequently found in the cross-phenotype analysis. Dot colour represents pleiotropy, i.e. the number of phenotype pairs showing (anti-)coherent signal for a given gene. Dot sizes are inversely proportional to *p*-values. Obtained using *PascalX* gene and cross-scoring respectively[57] *PascalX*, *p*-values are based on a two-sided Chi-square test and were corrected for multiple testing using the Bonferroni method (significance threshold set to 0.05/number of tested genes).

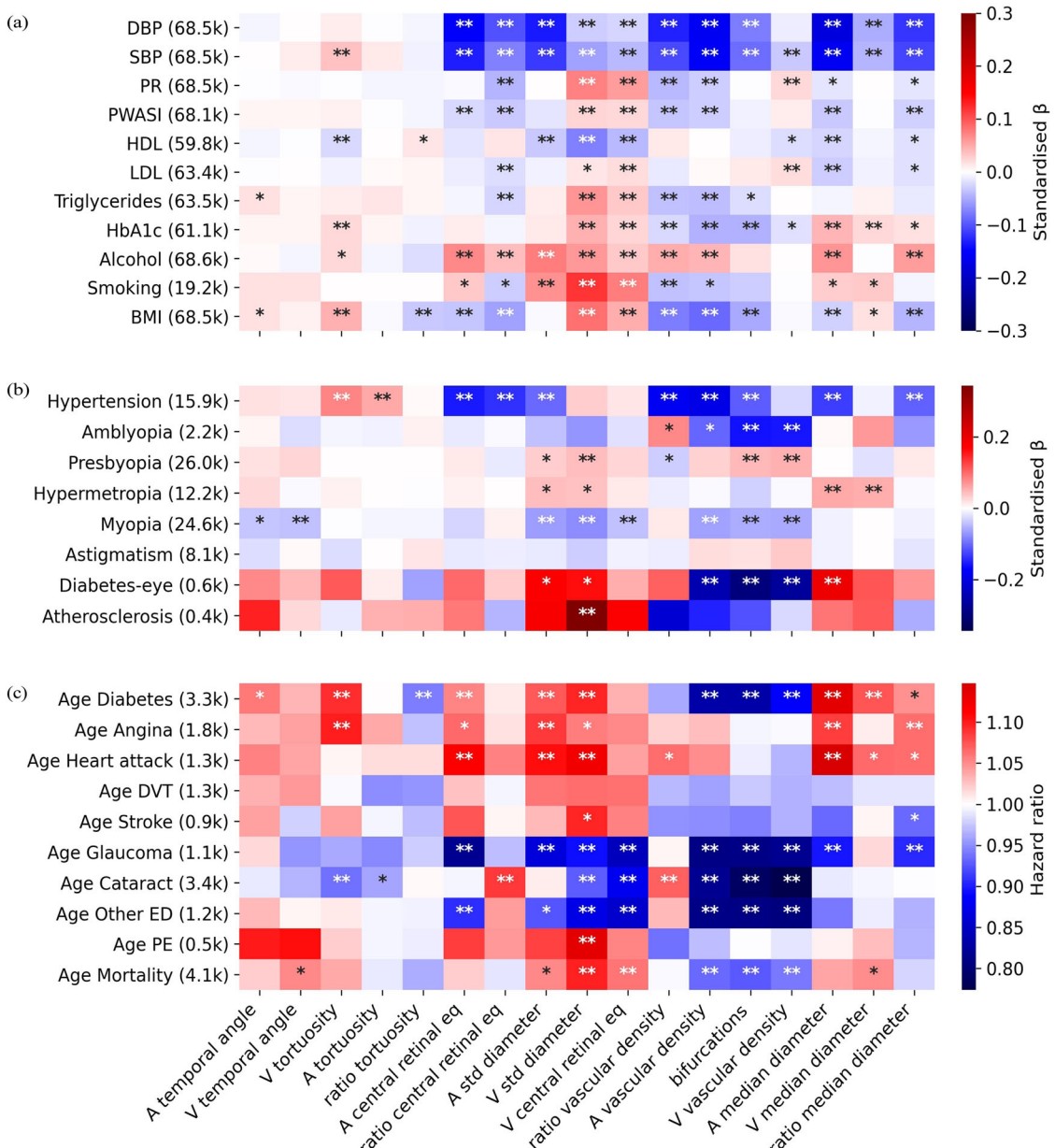

**Fig. 4 | Phenotypic association of IDPs with risk factors and diseases.** The *x*-axis shows IDPs and the *y*-axis shows risk factors and diseases. The numbers in parentheses correspond to the number of subjects with this information for which we were able to measure at least one of the 17 IDPs, for continuous diseases. For binary disease states, it represents the number of subjects who were cases and had data for at least one of the 17 IDPs. Linear (**a**) and logistic (**b**) regressions were used for continuous and binary disease states, respectively. For age-of-death and other severe diseases with the age-of-onset information, Cox proportional hazards regression was performed (**c**). In all models, phenotypes were corrected for age, sex, eye geometry, batch effects, and ethnicity. The colour indicates standardized effect sizes for linear and logistic regressions or hazard ratios for Cox models. Asterisks indicate the level of statistical significance ($^*p < 0.05/N_{tests}$, $^{**}p < 0.001/N_{tests}$, where $N_{tests} = N_{IDPs} \times N_{traits}$, and $N_{traits}$ is the number of diseases or risk traits considered in each panel). Labels: PR Pulse rate, PWASI Pulse wave arterial stiffness index, HDL High-density lipoprotein, LDL Low-density lipoprotein, HbA1c Glycated haemoglobin, Alcohol Alcohol intake frequency, Smoking pack-years, BMI Body mass index, Diabetes-eye Diabetes related to the eye, DVT Deep vein thrombosis, Other ED: all types of severe eye diseases not included explicitly, PE Pulmonary embolism.

of bifurcations, $\beta \in [-0.30, -0.25]$, while atherosclerosis was positively associated with venous diameter variability, $\beta = 0.34$ (Fig. 4b).

The Cox model analysis for age-of-onset phenotypes included severe eye- and cardiovascular diseases, diabetes, and age at death. Both vascular densities displayed consistent associations with the age of diagnosis for all eye-related diseases. Diabetes age-at-diagnosis shared similar associations with our vascular IDPs as retinal diabetes, but was also associated with venous tortuosity and central retinal arterial equivalent, among others. Heart attack was associated with larger

median arterial diameter and central retinal arterial equivalent, as well as diameter variability in both vessel types. Importantly, earlier deaths were most strongly associated with increased venous diameter variability and less strongly with increased central venous retinal equivalent, increased venous temporal angles, decreased vascular density, and fewer bifurcations. In general, venous diameter variability was associated with almost all diseases, including the only associations with pulmonary embolism and stroke (Fig. 4c). The complete table of the standardized effect sizes, hazard ratios and *p*-values is available on Figshare.

While the vascular densities and the number of bifurcations were highly inter-correlated, their associations with diseases sometimes differed. For instance, SBP and hypertension were associated with arterial vascular density and the number of bifurcations but not with venous vascular density. An equivalent analysis of 17 leading principal components (PCs) of our IDPs revealed no additional or stronger disease associations compared to the raw IDPs (see Supplementary Note 2).

### Genetic associations with diseases and causality analysis

To investigate the extent of common genetic architectures between the vascular IDPs and risk factors, we first used LDSR to estimate their genetic cross-correlations. Notably, we observed the strongest negative correlations ($\approx-0.30$) between BP measures and the ratios of the central retinal equivalents, and vascular densities. Body mass index (BMI) was correlated positively with median vein diameter ($\approx0.15$), and its variation ($\approx0.18$), and negatively with arterial vascular density and the number of bifurcations (Fig. 5a). HbA1c was also negatively correlated with the number of bifurcations, while HDL was negatively

correlated with venous diameter variability, but only with marginal significance. For binary disease states, we found weaker associations; however, hypertension exhibited associations similar to those of BP. For further details on binary disease states, see Supplementary Note 3.

To compare the phenotypic and genetic associations between vascular IDPs and risk factors, we plotted the effect sizes of the phenotypic linear regressions against genetic correlations for each IDP/risk factor pair (Fig. 5b). The slopes of the best-fitting lines through the corresponding points were positive for all risk factors, except for LDL cholesterol.

A strong genetic association between two phenotypes does not imply the existence of a causal link. To systematically assess potential causal relationships between vascular IDPs and risk factors, we performed bidirectional two-sample Mendelian Randomisation (MR) analyses[58,59]. Causal estimates were derived from the inverse variance weighted method using the TwoSampleMR R package[60,61]. We used false discovery rate (FDR) to correct for multiple testing and we also performed sensitivity analyses that confirmed the robustness of our

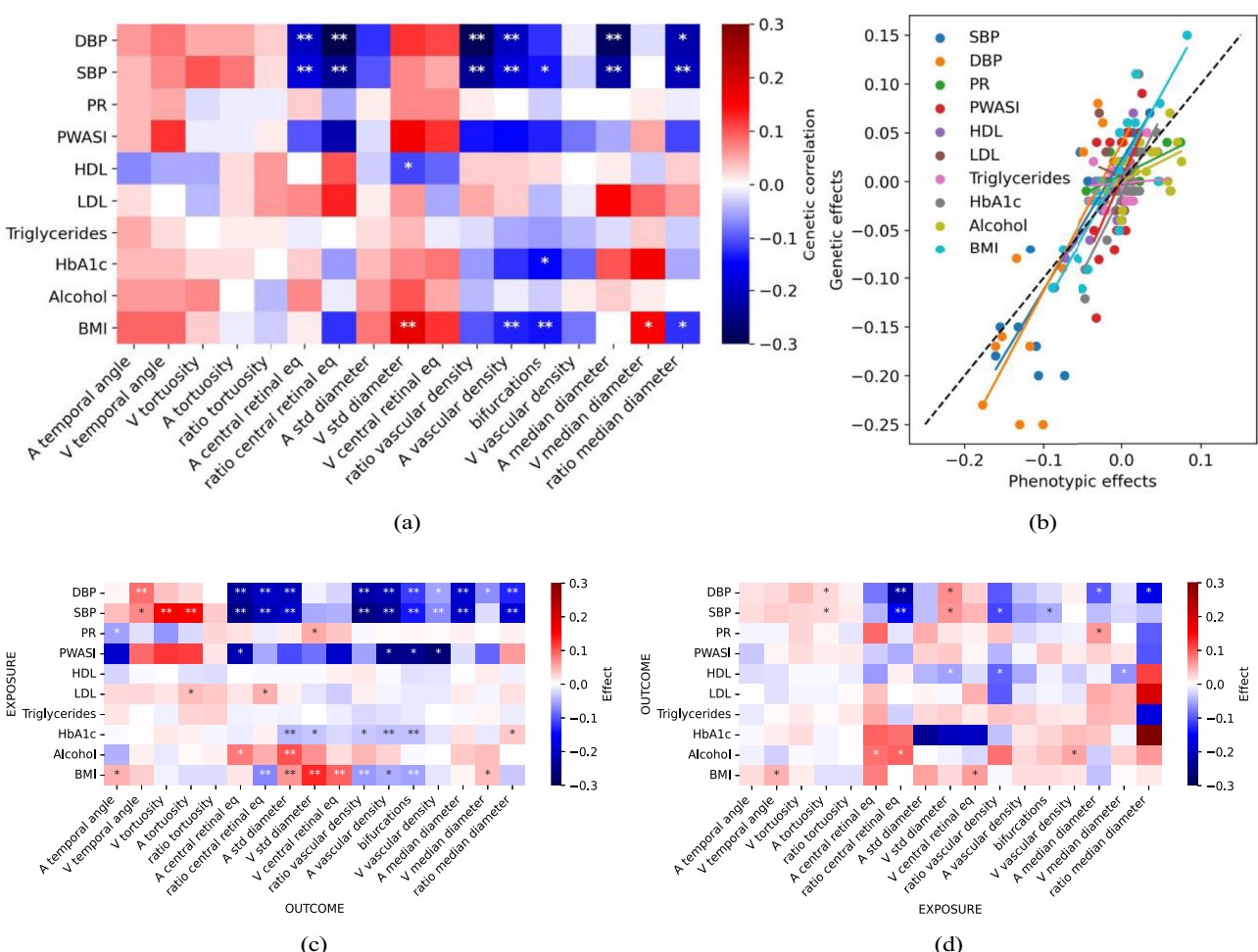

**Fig. 5 | Genetic correlations and causal effect estimates between IDPs and risk factors. a** Genetic correlation between IDPs and risk factors, computed using LDSR[55]. UKBB sample sizes are given in the 'N GWAS' column in Supplementary Table 2, and corresponding disease sample sizes are described on the Neale lab website (see Methods). The colour indicates the genetic correlation coefficient and the asterisks indicate the level of statistical significance (*$p < 0.05/N_{test}$, **$p < 0.001/N_{test}$, being $N_{test} = N_{IDP\,s} \times N_{LinearDiseases}$). **b** Correlation between phenotypic and genetic correlations of IDPs with risk factors. **c** Causal effect estimates with risk factors as exposures and IDPs as outcomes. **d** Causal effect estimates with IDPs as exposures and risk factors as outcomes. The colour indicates the causal effect

estimates based on the F statistic of the inverse variance-weighted MR method. The level of statistical significance is indicated with a single asterisk for nominal significance without correction for multiple testing (*$p_{uncorrected} < 0.05$) and two asterisks for a FDR (**$p_{FDR} < 0.05$). Risk factor genetic sample sizes: DBP and SBP: 340 k; High BP: 360 k; PR: 340 k; Pulse wave ASI: 118 k; HDL cholesterol: 315 k; LDL direct: 344 k; Triglycerides: 344 k; HbA1c: 344 k; Alcohol intake frequency: 361 k; Smoking status: 360 k; BMI: 360 k. And, IDPs genetic sample sizes: A temporal angle: 55 k; V temporal angle: 58 k; A/V and ratio tortuosity: 68 k; A, V and ratio central retinal eq: 65 k; A/V std diameter and bifurcations: 68 k; A/V and ratio vascular density: 68 k; A/V and ratio median diameter: 68 k.

results (see Methods). Full MR risk factors (IVW) are available on Figshare.

　　Using risk factors as exposures, we observed evidence for causal effects on most of the IDPs, even after adjusting for multiple testing (Fig. 5c). Notably, we found strong evidence for negative effects of DBP and SBP on many different vascular IDPs, including arterial (and ratio) central retinal equivalent, arterial (and ratio) median diameter, vascular density and bifurcations. Also, SBP had a positive effect on arterial and venous tortuosity. BMI had a positive effect on venous (and arterial) diameter variability and venous central retinal equivalent, while it showed a negative effect on ratio central equivalent, ratio vascular density and bifurcations. Furthermore, alcohol intake had a positive effect on arterial (but not venous) diameter variability, while HbA1c levels had a weak negative effect on arterial vascular density, bifurcations and arterial diameter variability. We also observed other potential causal effects, but only with marginal significance ($p_{uncorrected} < 0.05$). For example, PR had a positive effect on venous diameter variability and a negative effect on arterial temporal angle, while LDL levels had a positive effect on arterial tortuosity and ratio central equivalent.

　　In contrast, using IDPs as exposures, only the causal effect of ratio central equivalent on BP (SBP and DPB) survived FDR correction. We also observed potential causal effects on risk factors for many other vascular traits, but only with marginal significance ($p_{uncorrected} < 0.05$) (Fig. 5d). For example, we identified arterial tortuosity and venous diameter variability as positive causal factors for BP, while the ratio of central equivalent had a negative effect on BP. Venous diameter variability (and venous median diameter) also had a negative effect on HDL levels, while arterial median diameter had a positive effect on PR. Significant causal effects were also found between vascular IDPs and binary disease states; see Supplementary Note 4.

　　To identify genes that were jointly associated with IDPs and risk factors, we first computed simple intersections between the corresponding gene sets of such pairs (Fig. 6a). This revealed some sizable overlaps, in particular for vein diameter variability and central equivalent with BP measures, triglycerides, HbA1c, and BMI. Applying *PascalX* cross-GWAS analysis[57], as previously for the pairs of two vascular IDPs, we were able to identify many additional candidates for pleiotropic genes (Fig. 6b, c). We observed that the sets of coherent genes (Fig. 6b) tended to be largest for phenotype pairs whose sets of associated genes already had a sizable overlap, while we found large sets of anti-coherent genes (Fig. 6c) for some phenotype pairs whose associated genes had no or only little overlap, notably for DBP and the ratio of the central retinal equivalents, the ratio of the vascular densities and the median arterial diameter and its variability. Moreover, the ratio of central retinal equivalents shared multiple anti-coherent genes with BP and BMI. All PascalX IDP-disease cross gene scores are available on Figshare. For a similar analysis for binary disease states, we found weaker associations; however, hypertension exhibited associations similar to those of blood pressure. For further details on binary disease states, see Supplementary Note 3.

　　Shared pathways between the IDPs and diseases can be found in Supplementary Note 5, and on Figshare.

**Replication analysis**

We had access to CFIs from two independent smaller cohorts, namely the Rotterdam Study (RS, $N = 8\,142$ participants) and OphtalmoLaus ($N = 2\,276$ participants), from which we computed our 17 IDPs. While we used an identical analysis pipeline for OphtalmoLaus, a specialized annotation software was developed for the RS images, adapted to their specifics, notably including a dedicated and internally validated vessel segmentation tool ("VascX", manuscript in preparation). Phenotypic correlations $r^{(p)}$ between IDPs obtained in the two replication cohorts were globally concordant with those from the UKBB data ($\rho = 0.86$, $p = 7.5 \times 10^{-42}$ for OphtalmoLaus and $\rho = 0.69$, $p = 1.8 \times 10^{-20}$ for RS, see

Fig. 7a, and Supplementary Note 6). Notably, we replicated the high phenotypic correlation between vascular densities and the number of bifurcations and the low correlation between the temporal angles and the other IDPs. The main difference was that for RS data we observed a positive correlation between the variability in arterial diameter and vascular densities, while it was negative for UKBB and OphtalmoLaus, which may be due to the different vessel segmentation tool used for RS images. We also observed many consistent associations between IDPs and disease traits in the RS (see Supplementary Figs. 8–11).

　　Because of its larger sample size we used exclusively RS data to attempt replication of our genetic associations results in the UKBB. Since RS consists of four sub-studies with different sample sizes (RS-I: 2 391, RS-II: 877 RS-III: 2 811, and RS-IV: 2 063 participants) GWAS were performed independently for each of them, and then meta-analysed (see Supplementary Figs. 12 and 13 for the corresponding Manhattan and QQ-plots). Applying LDSR to estimate IDP SNP heritabilities and genetic cross-trait correlations, we observed overall consistency with the estimates from the UKBB data ($\rho = 0.74$, $p = 0.001$ for the former and $\rho = 0.45$, $p = 5.63 \times 10^{-12}$ for the latter, see Fig. 7b, c). Notably, in RS arterial tortuosity ($h^2 = 0.22 \pm 0.07$), the tortuosity ratio ($h^2 = 0.20 \pm 0.06$) and variability in venous diameter ($h^2 = 0.18 \pm 0.06$) also had the highest heritability estimates. Plotting the effect sizes for significant SNPs in the UKBB against those of the RS (Fig. 7d) revealed highly significant correlations and concordance in direction for the vast majority of IDPs. We then sought to replicate individual genetic associations. To this end, we applied the well-established Benjamini-Hochberg procedure[62]. With a fixed FDR of 0.05, we replicated 86 SNPs out of 195 across the four IDPs shown in Fig. 7e, and 232 SNPs out of 566 were replicated across the 17 main IDPs (see Supplementary Note 7 and Supplementary Table 3).

　　Finally, to identify the genes modulating specific vascular IDPs in the replication cohort (RS), we employed *PascalX*[56,57]. We applied the Benjamini-Hochberg procedure[62] for the genes. With a fixed FDR of 0.05, we replicated 93 genes out of 310 across the four IDPs shown in Fig. 7f. The complete set of RS gene scores are available on Figshare, and a table with genes associated with each RS IDP can be found in Supplementary Table 4.

## Discussion

In this study, we established an automated analysis pipeline to extract 17 retinal vascular phenotypes from CFIs and applied it to over 130 k CFIs of close to 72 k UKBB subjects. While some of these phenotypes had previously been studied individually, our work provides a common reference. Our phenotyping procedure, automated and open access, enabled us to study jointly a large panel of retinal vascular phenotypes, some of which (temporal angles, central equivalents, number of bifurcations) had not been assessed previously in a large cohort. We provided a comparison of the phenotypic with genotypic correlation structures of these IDPs. We estimated their heritabilities, and elucidated associated genes and pathways, allowing us to identify common and disjoint genetic architectures. We studied associations of our IDPs with a spectrum of diseases and risk factors providing evidence of their complementarity for indicating specific disease risks. For validation, we reproduced the phenotypic correlation structure in two independent cohorts, the RS and OphtalmoLaus, with 8.1 k and 2.2 k participants respectively, and validated numerous GWAS results in the RS. Importantly, the RS analysis pipeline was coded independently (to adapt to the specifics of their CFIs), such that these successful replication results provide strong evidence that our phenotyping is robust and not driven by cohort specific effects.

　　The phenotypic correlations revealed distinct clusters of vascular IDPs. One cluster includes vascular arterial and venous density, previously linked to fractal dimension[51], and the number of bifurcations. Our results, both the discovery and replications, suggest that the number of bifurcations, which is challenging to identify, can be reliably

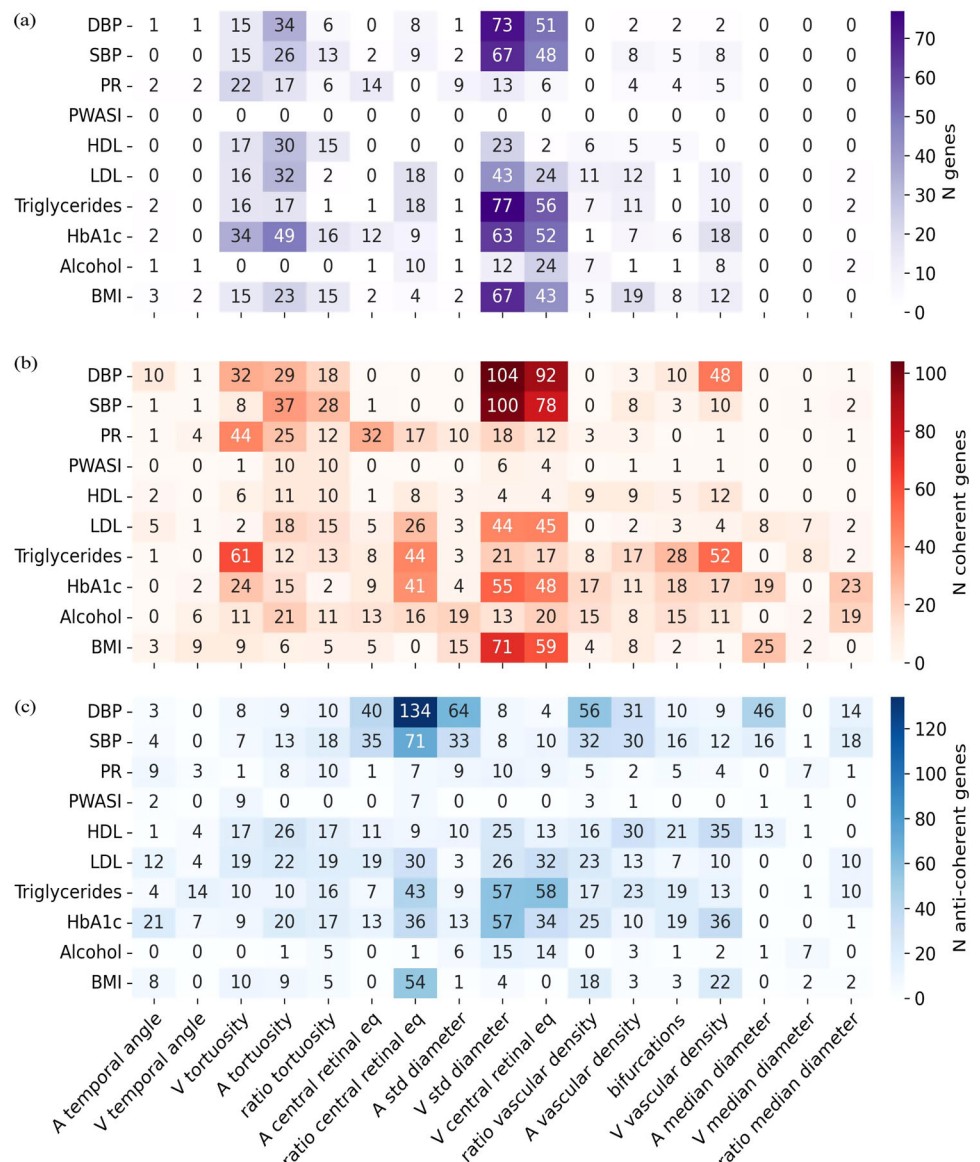

**Fig. 6 | Number of genes shared between IDPs and risk factors. a** Gene-scoring plain intersection. Each cell shows the number of intersected genes in phenotype pairs. **b, c** Cross-phenotype coherence analysis showing the number of coherent (**b**) and anti-coherent (**c**) genes between phenotype pairs. Summary statistics for risk factors were obtained from http://www.nealelab.is/uk-biobank (see for risk factor sample sizes). Gene-level p-values were derived from two-sided Chi-square test statistics using *PascalX*[57] and corrected for multiple testing with the Bonferroni method (significance threshold set to 0.05/number of tested genes). Risk factor genetic sample sizes: DBP and SBP: 340 k; High BP: 360 k; PR: 340 k; Pulse wave ASI: 118 k; HDL cholesterol: 315 k; LDL direct: 344 k; Triglycerides: 344 k; HbA1c: 344 k; Alcohol intake frequency: 361 k; Smoking status: 360 k; BMI: 360 k. And, IDPs genetic sample sizes: A temporal angle: 55 k; V temporal angle: 58 k; A/V and ratio tortuosity: 68 k; A, V and ratio central retinal eq: 65 k; A/V std diameter and bifurcations: 68 k; A/V and ratio vascular density: 68 k; A/V and ratio median diameter: 68 k. Retinal IDP sample sizes are listed in the 'N GWAS' column in Supplementary Table 2.

estimated using vascular densities. Interestingly, diameter variabilities showed stronger correlations with central retinal equivalents (of the same vessel type) than with the median diameters, suggesting that larger vessels dominate diameter variability. Indeed, median diameters (especially for veins) are confounded by vascular density, which is probably due to CFIs with lower vascular density exhibiting less blood vessels of small caliber. The latter may be due to degeneration of the vasculature, but also to poorer image quality, in particular blurriness, which is impacted by corneal opacity. More work is needed to disentangle the different distributions to establish robust diameter measures for blood vessels not proximal to the OD. Finally, the temporal angles exhibited little correlation with other vascular IDPs, both in the discovery and replication cohorts, indicating that they may be influenced by non-vascular factors, such as eye anatomy.

The genetic correlation between IDPs largely mirrored their phenotypic correlation, supporting Cheverud's conjecture which states that phenotypic correlation can be used as a proxy for genetic correlation[63]. However, in our study genetic correlations were slightly larger on average, indicating potential differential effects of environmental factors on correlated IDPs.

We observed significant variation in heritability estimates among different vascular IDPs. Tortuosity and vascular density showed comparatively high heritability, consistent with some previous findings[27,51]. A recent study[26] estimated heritability of retinal arteriolar tortuosity at 0.51 using UKBB data, substantially higher than our 0.25 estimate. Our point estimate in the RS is 0.22 ± 0.07, which is consistent with our estimate from UKBB data. We note that here we used median tortuosity across all vessel segments independent of their caliber and

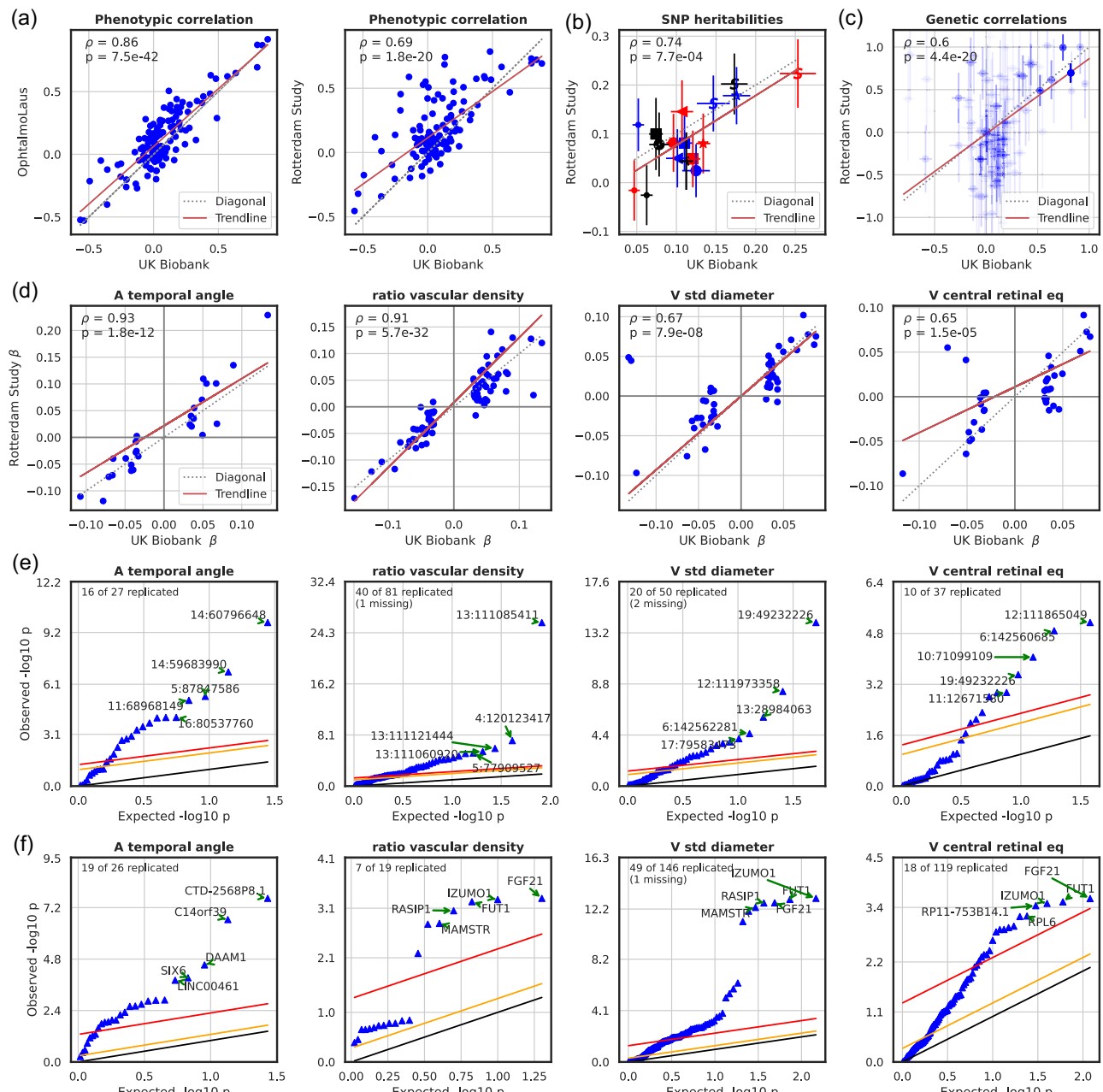

**Fig. 7 | Phenotypic and genetic replication of IDPs.** UKBB sample sizes are given in Supplementary Table 2, RS sample size is 8 142, OphtalmoLaus sample size is 2 276. **a** Scatter plot of the phenotypic correlations between our 17 IDPs in the UKBB and in the replication cohorts, OphtalmoLaus (left) and RS (right). IDPs were corrected for age, sex, eye geometry, and ethnicity (see Methods). Correlations of correlations and their corresponding *p*-values are displayed. **b** Correlation of SNP heritabilities, using LDSR (see Methods for statistical test), between our 17 IDPs in the discovery (UKBB) and the replication cohort (RS). **c** Scatter plot of the genetic correlations, using LDSR (see ref. 54 for statistical test), between our 17 IDPs in the discovery (UKBB) and the replication cohort (RS). Weighted-least square regression was used to determine trendline and the significance of the association. To distinguish between the different IDPs, the following colour and shape legend was utilized: 'S' denoted tortuosity, '*' for standard deviations, '◁' for temporal angles,

the '≺' for bifurcations, and '□' for vascular density. While the red colour is used for arteries, blue for veins, and black for no specific vessel type. Genetic correlations are measured as the correlation of their effect sizes across genetic variants, accounting for linkage disequilibrium (LD) (see ref. 54). **d** Correlation of effect sizes at the SNP level in the discovery (UKBB) and the replication cohort (RS), based on OLS of SNP genotype onto phenotype value. **e** Benjamini-Hochberg procedure on discovery lead SNPs from the UKBB using the RS. FDR = 0.05 in red, FDR = 0.5 in orange, and observed = expected line in black. The label "missing" indicates that these SNPs were not available in the replication cohort. *P*-values are based on OLS of SNP genotype onto phenotype values. **f** Benjamini-Hochberg procedure on genes discovered in the UKBB using the RS. The colour code is the same as in the previous subfigure. The complete figures can be found in Supplementary Note 7. *P*-values are based on *PascalX*'s two-sided Chi-square test statistic.

length, even though tortuosity may vary as a function of blood vessel length and diameter, calling for more refined or stratified measures.

Notably, venous diameter variability also exhibited high heritability, which may be partially attributed to vascular beading, severe forms of which are known to be inherited[64]. Interestingly, the heritability of temporal angles was relatively low, likely reflecting sensitivity

to environmental factors, as well as measurement noise and methodological variations, such as variability in the position of the OD across images, and variation due to refraction that may be insufficiently corrected by spherical and cylindrical power as our covariates. Future work could use heritability as a guide for developing more robust measures of these angles both in terms of the methodology for

extracting them (see Supplementary Methods "Phenotype extraction" for our procedure) and implementing proper corrections for potential confounders. Finally, median vessel diameters obtained the smallest heritabilities, consistent with their known dependence on the environment, such as vasoconstriction induced by stress, and diseases, as well as the aforementioned confounding by image quality and age-dependent reduction in detectable small blood vessels.

Gene analysis of individual IDPs replicated some previously identified gene associations with explicit vascular phenotypes, namely tortuosity[26–28], fractal dimension, vascular density[51], and vessel width[26], as well as DL vascular phenotypes representing the latent space of an autoencoder[43] (see Supplementary Discussion "Comparison with previous GWAS" for complete list). Diameter variability and central retinal equivalent, particularly for veins, received numerous gene associations, providing evidence for the genetic complexity of these phenotypes. Finally, the ratios of vascular measures, combining arterial and venous components, showed associations with genes not found in individual measures, indicating sensitivity to vessel-type specific effects. Generally, phenotypes with higher heritability tended to have more associated genes, with some exceptions like arterial diameter variability.

No single gene was significantly associated with all the vascular IDPs. The most frequently observed gene was *LINC00461*, also known as visual cortex-expressed gene (VISC), an evolutionarily conserved long non-coding RNA that produces several alternatively spliced transcripts[65]. Other frequently observed genes are related to eye diseases (*PDE6G* has been linked to Retinitis Pigmentosa, and *SIX6* to glaucoma, myopia, and retinal degeneration) or vascular processes (*FLT1* linked to hypertension and heart disease). Also, *HERC2* and *OCA2*, two neighbouring genes related to pigmentation, were linked to multiple IDPs. *OCA2* is involved in the production of melanosomes, and therefore directly contributes to the formation of pigments, while mutations in *HERC2* have been found to modulate *OCA2* expression[66]. We speculate that less eye pigmentation leads to reduced protection from damaging light, which could cause global changes on the retinal surface, including vascular morphology.

Our study of phenotypic associations between vascular IDPs and disease-relevant phenotypes confirmed some previously known associations, such as the link between vessel diameter and hypertension, likely due to alterations in vascular resistance and blood flow[3,4,67]. Interestingly our MR analysis provides support for a low ratio of the central retinal equivalent being causal for high BP. This is consistent with a positive (but not significant) phenotypic and genetic correlation between central retinal venular equivalent and hypertension/BP. We also observed significant associations between IDPs capturing arterial vessel caliber and heart attack, consistent with previous findings[18,48–50,68,69], and the previously observed link between tortuosity and cardiovascular risk factors[10,70]. For a more complete discussion refer to Supplementary Discussions "Previous associations with diseases", and "Replication of previously identified associations of retinal vascular phenotypes with diseases".

Additionally, we found diameter variability in both vessel types to be positively associated with heart attack, possibly due to the formation of plaques that enlarge the vessel while reducing lumen diameter and negatively impacting blood flow. While atherosclerosis has a similar association profile across the vascular phenotypes, interestingly the association with venous diameter variability was the strongest and the only one passing statistical significance. This signal may be driven by artery-vein crossings and calls for further investigation.

Type-2 diabetes is known to affect the microvascular circulation in the retina resulting in a range of structural changes unique to this tissue, such as neovascularization[71], which can affect vascular density[51] and arteriolar tortuosity[72], among others. We note that the association profile of our IDPs with HbA1c, which is used to diagnose type-2

diabetes[73], is similar, but typically a bit weaker in terms of effect size and significance than the latter, suggesting that our Cox model taking into account the age of onset of the disease endpoint has more power. Given that diabetes is a known risk factor for cardiovascular disease, including angina, this could explain why many associations between type-2 diabetes and IDPs related to vessel caliber are also observed for angina and heart attack.

Age at death is the only trait, besides myopia, that is associated with the venous temporal angle. Moreover, age at death also shares some of the associations of specific diseases with our IDPs, such as a negative association with the number of bifurcations (in line with the results for hypertension and type-2 diabetes reported above), and positive associations with several vein-related phenotypes besides the temporal angle, including the diameter variability, the median diameter (in line with the results for stroke, type-2 diabetes, heart attack and atherosclerosis reported above). It seems plausible that the reduced lifespan for these common diseases explains the observed associations.

MR analysis allowed assessment of potential causal directions for the links observed in the correlation analysis. Overall the effects of CVD risk factors on our vascular IDPs tend to be stronger and more significant than the reverse, underlining the usefulness of IDPs for early diagnosis of CVD. Consistent with previous findings, we found that individuals with genetically elevated BP tend to have lower retinal vascular density[51]. In addition, we found several other IDPs being affected, suggesting that vascular remodelling can be caused by elevated BP. Also, elevated BMI may cause higher variability in venous and arterial diameter, consistent with the finding that obesity may decrease venous return of blood from the lower extremities thereby increasing the risk of chronic venous insufficiency[74]. Amongst the strongest potential reverse causal effects is the aforementioned decrease in BP due to a higher ratio of the central retinal equivalent. Additionally, we found that higher arterial tortuosity tends to increase BP, confirming previous findings[26]. Interestingly, we also found that vein diameter variability may also cause higher BP, which could be mediated by anatomical alterations such as venous beading. However, those findings were not significant after correcting for multiple testing. The recent paper by Jiang et al.[26] also reported a causal effect of arterial tortuosity on Coronary Heart Disease (CHD). We were able to confirm this causal link of arterial tortuosity on CHD, while no other retinal trait was found to be causal (see Supplementary Note 4).

In the gene analysis, we observed that the pairs of diseases and IDPs with the highest positive correlations tended to have a greater number of shared genes, both in the intersection and coherence analyses (see Supplementary Fig. 7 "LDSR genetic correlation against PascalX"). It is worth noting that this was not limited to pairs with high-significance values. Besides, we again observed that the patterns differed in the analysis of coherent and anti-coherent genes. For example, vein diameter variability and BMI had multiple genes associated coherently, implying that the genes modulating them acted in the same direction. Conversely, between BMI and the central equivalent ratio, the majority of genetic effects acted in the opposite direction. This is consistent with the previous finding that vein diameter variability and ratio central equivalent shared many anti-coherent genes.

While our study pushes the boundaries of analysing retinal vascular phenotypes, it has several important limitations: First, limiting our study to data from the UKBB, RS, and OphtalmoLaus makes our findings specific to a population of mostly European ancestry. Second, there are some other potentially relevant vascular IDPs that we did not analyse, including branching angles, artery-vein crossings, and neovascularization. Third, our GWAS did not include the analysis of sex chromosomes or rare variants. The exclusion of these factors may have limited the scope of the study and prevented the identification of potential associations between genetic variations and the

development of certain diseases or phenotypes. Fourth, the variability in the number of cases with severe diseases, as shown in Fig. 4, may influence the results. Finally, summary statistics used for the genetic analysis of binary disease states were not obtained using GWAS with logistic regression, but linear regression, which can have an effect a sensitivity effect on their results (see methods: Genetic association with diseases).

In summary, this study establishes a common framework for studying multiple vascular phenotypes of the retina. The explicit characterisation of retinal vasculatures will be useful both for clinical research further exploring their usefulness as biomarkers for systemic diseases, and fundamental research, where we provide an important alternative reference to implicit characterizations of the retina, such as the recent "Foundation model" for retinal images[44]. Our analysis of genes and pathways unveiled a strikingly limited intersection, indicating that the mechanisms governing these phenotypes are largely independent. Our findings regarding the association between disease phenotypes affirmed some established knowledge while uncovering numerous additional connections. Specifically, we observed a plethora of additional links between diameter variability, particularly in veins, and various disease phenotypes such as age of mortality, pulmonary embolism, and myocardial infarction. While more work is needed to further validate and extend our findings, our analyses provide evidence that we start to have sufficient power for obtaining functional, as well as some initial causal insights into the genetic and disease-related processes modulating the retinal vasculature.

## Methods

### UK Biobank data
The UKBB is a population-based cohort of ~488k subjects with rich, longitudinal phenotypic data including a complete medical history, and a median 10-year follow-up[75,76]. Standard retinal 45° CFIs were captured using a Topcon 3D-OCT 1000 Mark II. 173 814 images from 84 813 individuals, were analysed. Genotyping was performed on Axiom arrays for a total of 805 426 markers, from which ~96 million genotypes were imputed. We used the subset of 15 599 830 SNPs that had been assigned a rsID. Baseline and disease information about the subjects whose images were analysed can be found in Supplementary Tables 2 and 5.

### Image segmentation and quality control
Raw CFIs were segmented into pixel-wise segmentations of arteries and veins using the DL model LWNET[40], which had been trained and validated on the publicly available DRIVE dataset. Briefly, the LWNET consists of two concatenated small U-Nets, allowing for faster training. To detect the OD, a random set of 100 CFIs of varying quality from the UKBB were first annotated, and the resulting ground truths were then used to retrain a standard U-Net previously trained to detect the OD in various public datasets[41]. Last, branch points of vessels and vessel-segment-wise and centerlines were extracted using skeletonization, and diameters were extracted using the distance transform, both provided in ARIA[77].

A published QC method[51] to assess image quality in the UKBB was used, and the 75% highest-quality CFIs according to this method were retained for further analysis. In this method, the image quality of 1000 CFIs was quantified by professional graders, and a CNN was then trained to imitate the graders' quality assessment. A significant negative correlation between image quality and age was observed, $r = -0.21$, but was not corrected. For more information refer to Supplementary Methods. Additionally, to see how the threshold on the QC can affect the results refer to Supplementary Figs. 1–3.

### Phenotyping
Retinal vessel morphology was broadly phenotyped, drawing on a set of known relevant ophthalmological phenotypes, including a few that

were previously undescribed, such as diameter variability. Due to the lack of consensus definitions and methods for their measurements, their implementation can vary. Therefore, we implemented different definitions and methods for most phenotypes, see Supplementary Methods "Phenotype extraction" and Supplementary Methods "Correlation structure and heritability of extended list of retinal vascular IDPs". In this study, we focused on 17 representative phenotypes with significant heritability and relevant disease associations. These phenotypes were selected for independence, removing highly redundant phenotypes with similar definitions, but keeping some highly correlated phenotypes when they resulted from significantly different definitions. The mean between left and right eye measurements was taken whenever measurements from both eyes passed quality control, otherwise, the single eye measurement was used. Measurements from only the first measured time point were used whenever measurements from multiple time points were present. Cohen's d, $(\bar{r}_g - \bar{r}_p)/((SD_g^2 - SD_p^2)/2)^{0.5}$, was used to quantify the mean difference between genetic and phenotypic correlations.

We used two validation procedures: (1) visual inspection of random images in the UKBB to identify and refine potential confounders affecting certain phenotypes, and (2) the use of the DRIVE dataset as a proxy to assess performance in other datasets with similar characteristics, using ground truth on it for the number of bifurcations and temporal angles (Supplementary Methods "Validation of retinal vascular IDPs"). The measured DRIVE temporal angles are available on Figshare.

### Correction for covariates
IDPs were corrected for sex, age, age-squared, sex-by-age, sex-by-age-squared, spherical power, spherical power-squared, cylindrical power, cylindrical power-squared, instance, assessment centre, genotype measurement batch, and genomic PCs 1–20. Their associations with each phenotype have been visualized in Supplementary Fig. 14 "UKBB Covariates effects". For GWAS, raw phenotypes were transformed with the rank-based inverse normal transformation (rb-INT) before correction.

### Disease association
The list of diseases analysed includes vascular and eye-related diseases, risk factors, mortality, and other conditions previously found to be associated with the retina vascular system. The disease data were collected from the UKBB, and the official datafield identifier corresponding to each disease can be found in Supplementary Table 6, and on Figshare.

Different regression models were employed based on the nature of the disease traits. For risk factors, ordinary least squares (OLS) linear regression was used to estimate standardized effects using the 'statsmod-els.formula.api' library in Python 3.8.13. For binary and categorical disease phenotypes, logistic regression was applied, using the logit function from the 'statsmodels.formula.api' library.

Prior to conducting the regression analyses, a pre-processing step was performed to address potential con-founding effects. Covariates were regressed out of retinal IDPs and the obtained residuals were then used as regressors in the linear/logistic regression analyses. For Cox models, the covariates were added again to the models to adjust for potential non-linear effects.

The regression models were fitted using the adjusted independent variables, the estimates of regression coefficients (betas), and their corresponding standard deviation (std), or odds ratios were obtained. To determine the significance of regression coefficients, p-values were computed and compared to predefined alpha thresholds (0.05 and 0.001), divided by the total number of tests conducted (i.e., the number of independent variables multiplied by the number of diseases analysed).

In cases where time-to-event data was available and accurate diagnosis was feasible, multivariate Cox proportional hazards regression was utilized. This approach allowed us to estimate hazard ratios for diseases with time-to-event outcomes and reliable diagnoses. 'coxph' from the R packages 'survival' and 'survminer' were used with default parameters.

## Genome-wide analyses

GWAS for UKBB data was performed using BGENIE[76]. SNP-wise heritabilities and genetic correlations between IDPs were derived using LDSR[55]. LDSR estimates SNP-wise heritabilities as the OLS slope regressing SNP LD scores on their respective mean mean chi-squared statistics. LDSR estimates gentic correlations as the shared genetic basis between two traits by assessing the correlation of their effect sizes across genetic variants, accounting for linkage disequilibrium (LD). Gene and pathway scores were computed using *PascalX*[57,78]. Both protein-coding genes and lincRNAs were scored using the approximate "saddle" method, taking into account all SNPs within a 50 kb window around each gene. All pathways available in MSigDB v7.2 were scored using *PascalX'* ranking mode, fusing and rescoring any co-occurring genes <100 kb apart. *PascalX* requires LD structure to accurately compute gene scores, which in our analyses was provided with the UK10K (hg19) reference panel. Gene-level cross-GWAS coherence test between IDP pairs and between IDPs and diseases or risk factors was computed using the *PascalX* cross-scoring *zsum* method, testing for both coherence and anti-coherence of GWAS signals. Variants with a minor allele frequency of at least 0.001 were considered. Correction for bias due to sample overlap was done using the intercept from pairwise LDSR genetic correlation. The significance threshold was set at 0.05 divided by the number of tested genes. GWAS top hits are available on Figshare.

## Genetic association with diseases

For the genetic correlation between IDPs and diseases, the summary statistics of the diseases in LDSR format were obtained from the Neale lab (nealelab-ldsc-sumstat-files). LDSR was computed for the diseases and the IDPs. We limited ourselves to diseases that were categorized as 'High confidence' by the Neale lab.

Regarding the genes shared between IDPs and diseases, we used the GWAS summary statistics for diseases (nealelab-sumstat-files). However, in this case, we applied some additional filters deleting rsid values that did not start with 'rs', *p*-values that were missing, and low confidence variants. The significant genes for each disease are listed on Figshare.

Binary disease states were included in the genetic analysis, however, their results require additional caution, since the GWAS summary statistics from the Neale lab used linear regression for all the disease phenotypes, which is not ideal for non-continuous response variables. It should be noted that the covariates used for the GWAS analysis of our IDPs and those used for the risk factors/diseases are almost, but not exactly, the same (age, sex, age-squared, age-by-sex, age-by-sex-squared, and first 20 PCs for risk factors/diseases).

## Mendelian Randomisation

To perform the bidirectional two-sample MR analyses we used the TwoSamplesMR package in R[60]. We first considered independent SNPs significantly ($p < 5 \times 10^{-8}$) associated with the exposure as genetic instruments, pruning SNPs with $r^2 > 0.001$ to a lead SNP according to LD estimates from the UK10K reference panel[79]. In cases where the number of instruments was below 10, we relaxed the selection threshold for instruments to $p < 10^{-6}$. In general, the number of instruments tended to be lower when using IDPs, rather than disease traits as exposures, which may be due, at least partially, to lower sample sizes. For this reason, when using IDPs as exposures we relaxed the selection threshold for instruments to $p < 10^{-6}$.

To assess instrument strength, we computed the F-statistic from the regression of the exposure on the instruments[80] defined as $F = (R^2 \times (N - k - 1))/((1 - R^2) \times k)$, where $R^2$ is the explained variance of the exposure by the instruments, $N$ is the sample size of the GWAS for the exposure, and $k$ is the number of instruments. For all the forward and reverse MR pairs, the F-statistics were >50 suggesting that the selected SNPs were suitable instruments.

Causal estimates were based on the inverse variance-weighted method (IVW)[61]. In particular, we used a fixed-effect model when having three or less instruments, and otherwise a random-effects model. To complement and enhance the reliability of the results, we applied additional methods, namely MR-Egger, weighted median, and weighted mode[81,82] MR. For all exposure-outcome pairs, the estimated causal effects were consistent in the direction across the four methods whenever significant. Differences in significance levels are likely because the power of these additional methods is smaller than that of the IVW method[83]. MR risk factors for all methods are available on Figshare.

Notably, MR-Egger intercepts of most of the associations were not significantly different from zero, suggesting that no significant pleiotropy was detected (see Figshare cor complete MR-Egger results). Lastly, leave-one-out (LOO) analyses showed that the estimates were not biased by any single SNP (see Figshare for complete LOO results). Overall, the sensitivity analyses confirmed the reliability of most of our putative causal effects in both directions.

Since the number of UKBB subjects for which we extracted IDPs was much smaller than the number of UKBB subjects used to study disease phenotypes, our analysis can still be considered a two-sample MR setup, and potential bias due to sample overlap is expected to be small and in direction of the null[84,85].

## Replication

The RS is a prospective population-based cohort study of people living in Ommoord, a district of the city of Rotterdam[86]. The RS consists of four cohorts, all of which were used in this replication. Each cohort was followed for multiple rounds of follow-up examinations every 4–5 years. Most of the patient visits in the RS involved the capture of CFIs on both eyes. Due to the multi-decade span of the RS, multiple devices, capture conditions and fields (macula and disc centred) are present in the dataset. In the RS, DNA extraction was performed using whole blood samples following standardized and previously described protocols[86]. Genotyping was performed using both the Infinium II HumanHap550(-Duo) (RS-I & RS-II) and 610-Quad Genotyping BeadChip (RS-I & RS-III; Illumina, San Diego, CA, USA). Imputation of markers was performed using the Haplotype Reference Consortium version 1.1 as the reference panel[87]. The CoLaus study, initiated in 2003 in Lausanne, Switzerland, involves over 6 700 volunteers aged 35 to 75. OphtalmoLaus, a segment of CoLaus, delves into ocular health. OphtalmoLaus CFIs were acquired with Topcon 2000 or Topcon triton. The number of initial images was 6 503, corresponding to 2 276 participants. See Supplementary Methods "Replication methods" for more details.

**Phenotyping and cross-correlations.** In this study we made use of RS imaging from recent rounds (RS-I-4, RS-II-4, RS-III-1, RS-IV-1) due to the generally higher quality and number of these images, even though in some cases this meant less participants. We quality-controlled the images by automatically filtering out images where the OD was near or out of the bounds of the CFI. Both disc- and macula-centred images were included. After QC, the number of participants with usable images per RS cohort was 2 710 (RS-I), 1 169 (RS-II), 3 350 (RS-III), and 2 658 (RS-IV). Most visits in the RS

captured multiple CFIs per participant. The average number of usable CFIs per subject varied between 2.96 (RS-I) and 7.75 (RS-IV), including both eyes. A participant's features were computed as the mean over all their images' features.

To accommodate the use of disc-centred images and the high variability in imaging conditions present in the RS, we made use of segmentation and feature extraction methods trained and tested on RS data. The 17 phenotypes computed for the main study were implemented in a fundus analysis software used in-house following the original implementation.

To compute the RS phenotypic cross-correlations in Fig. 7, IDPs were corrected for age, sex, eye geometry, imaging device (if multiple were used within the same cohort, dummy coded), and genomic PCs 1–10. Eye geometry was included as a combination of the spherical and cylindrical powers into one variable as spherical equivalent (spherical power + cylindrical power/2).

For OphtalmoLaus, in the case of multiple images for the same participant, we kept the one with the highest QC score. The number of participants after removing the images that failed the segmentation or IDPs computation, and after QC, was $N = 1\,715$ participants. If both eyes survived this screening, we averaged out the phenotypes of the two eyes, while if only one eye survived then we considered the phenotypes of that eye. After that, we cross-referenced with the sample file and the final number of subjects was $1\,435$. We corrected for the following covariates: age, sex, age-by-sex, age-squared, cylindrical and spherical powers, spherical-squared, cylindrical-squared, and genomic PCs 1–10.

**GWAS.** As imputation and QC of the four RS-cohorts were done separately, we also performed the GWAS analyses separately and then meta-analysed the results. For the initial GWAS analyses, we performed linear regressions using Plink 2.0[88]. The included covariates were the same as for the phenotypic correlation, i.e. age, sex, eye geometry, imaging device, and genomic PCs 1–10. We performed an inverse variance weighted fixed-effect meta-analysis, using METAL software.[89] *P*-values for the association results were calculated by using the *z*-statistic. The meta-analysed significant hits were pruned using Plink 2.0. Variants were considered independent if they were at least 500 kb apart and had an $R^2 < 0.1$.

### Reporting summary
Further information on research design is available in the Nature Portfolio Reporting Summary linked to this article.

## Data availability
Source Data used to generate all figures are available on Figshare (https://doi.org/10.6084/m9.figshare.26509756). GWAS summary statistics are available on Zenodo (https://zenodo.org/records/12779552) and on the GWAS Calatog (https://www.ebi.ac.uk/gwas). Phenotypic data are under restricted ac-cess. Our phenotypic data derived from images can be accessed through the respective cohort platforms: UKBB (https://www.ukbiobank.ac.uk/), RS (https://www.erasmusmc.nl/en/research/departments/epidemiology), and OphtalmoLaus (https://www.colaus-psycolaus.ch/autres-etudes/ophtalmolaus). The raw UKBB data are protected and not open access; however, they can be obtained upon project creation and acceptance. Similarly, replication data from the RS and OphtalmoLaus studies can be made available to researchers upon request through a data transfer agreement.

## Code availability
Code is available on Zenodo: https://zenodo.org/records/13347953.

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

## Acknowledgements

The authors thank Leah Böttger and Sacha Bors for their valuable comments on the manuscript. The authors thank the entire EyeNED team for their various contributions during the development of the algorithms used in the RS replication. The authors are grateful to the study participants and the staff from the UKBB, RS, and OphtalmoLaus. Funding: Supported by the Swiss National Science Foundation grant no. CRSII5 209510 for the "VascX" Sinergia project. Supported by the Swiss Personalized Health Network (2018DRI13) for the project "SOIN (Swiss Ophthalmic Imaging Network)". Supported by the Claire and Selma Kattenburg Foundation (Prix Kattenburg 2022). Supported by Oogfonds, Stichting voor Ooglijders, Stichting voor Blindenhulp, Henkes stichting, and Landelijke Stichting voor Blinden en Slechtzienden (LSBS). Additional support was given by Erasmus Medical Center, Erasmus University, Netherlands Organization for the Health Research and Development (ZonMw), the Research Institute for Diseases in the Elderly (RIDE), the Ministry of Education, Culture and Science, the Ministry for Health, Welfare and Sports, the European Commission (DG XII), and the Municipality of Rotterdam.

## Author contributions

S.O.V., M.J.B, and S.B. designed this study. M.J.B. performed QC and image preprocessing. S.O.V. extracted the number of bifurcations, temporal angles, central retinal equivalents, and diameter variability. M.J.B. extracted OD positions, vascular density, and fractal dimension. M.T. extracted median diameter and tortuosity. M.J.B. performed post-processing (covariate correction and normalization). GWAS was performed by M.J.B. and S.O.V. for UKBB data. M.J.B. estimated heritabilities and cross-phenotype correlations. S.O.V. and M.J.B. performed gene and pathway analysis, with DP support. Gene-level cross-phenotype analysis was done by O.T.. Linear and logistic regressions were done by S.O.V. and O.T., and Cox model analyses by M.J.B.. Cross-phenotype LDSR and gene analysis between retina phenotypes and diseases were done by S.O.V. and O.T.. F.H. helped S.O.V. with the validation of the angles and bifurcation phenotypes. I.I. performed MR analysis. J.D.V.Q. and V.A.V. performed RS replication under the supervision of W.R., B.L., and C.C.W.K.. I.M. and A.E. performed OphtalmoLaus replication under the supervision of M.T. and R.S.. S.B. supervised all analyses. S.O.V., M.J.B., O.T., I.I., M.T., J.D.V.Q., I.M., and S.B. wrote the manuscript. All other authors contributed to the writing.

## Competing interests

The authors declare no competing interests.

## Additional information

[1]Department of Computational Biology, University of Lausanne, Lausanne, Switzerland. [2]Swiss Institute of Bioinformatics, Lausanne, Switzerland. [3]Department of Ophthalmology, Erasmus MC University Medical Center, Rotterdam, The Netherlands. [4]Department of Epidemiology, Erasmus MC University Medical Center, Rotterdam, The Netherlands. [5]Department of Ophthalmology, University of Lausanne, Fondation Asile des Aveugles, Jules Gonin Eye Hospital, Lausanne, Switzerland. [6]Platform for Research in Ocular Imaging, Department of Ophthalmology, University of Lausanne, Fondation Asile des Aveugles, Jules Gonin Eye Hospital, Lausanne, Switzerland. [7]Department of Ophthalmology, Amsterdam University Medical Centres, Amsterdam, The Netherlands. [8]Department of Ophthalmology, Radboud University Medical Center, Nijmegen, The Netherlands. [9]Institute of Molecular and Clinical Ophthalmology, University of Basel, Basel, Switzerland. [10]Department of Integrative Biomedical Sciences, University of Cape Town, Cape Town, South Africa. [11]These authors contributed equally: Sofía Ortín Vela, Michael J. Beyeler. ✉e-mail: sofia.ortinvela@unil.ch; michael.beyeler@unil.ch; sven.bergmann@unil.ch

