## [Peer Review File · Nature Communications]

REVIEWER COMMENTS

Reviewer #1 (Remarks to the Author):

This paper describes an automated system to extract retinal vessel indices from colour fundus images (CFI), with application to 130K CFI from 72K UK Biobank participants. The paper outlines GWAS and phenotype associations with retinal vessel traits, which show heritability of between 5 to 25%, potentially providing causal insights into disease outcomes and processes.

Unfortunately, the paper has a number of limitations. A major concern is the lack of an independent replication data-set to confirm the findings, which is increasingly becoming standard within this field. In the absence of a validation data-set, there should be greater emphasis on previous work, and this only receives cursory mention, which is largely assigned to supplemental material. There is a need for more direct comparison with previous work in the main text paper to provide validity, i.e., to what degree does this work mirror earlier work, particularly in UK Biobank? In my view supplemental material #11 could be built-upon and usefully included in the paper, with clearer consideration of the overlap (or lack of overlap) with loci identified previously. It is noteworthy that recent work has shown higher heritability for some retinal vascular traits (particularly for retinal arteriolar tortuosity) than reported here. Reasons for these differences could be considered further, and the degree to which this might reflect methodological approaches. My view is that given recent papers published (especially in UK Biobank) the current work needs to clearly outline the added value of this work. While these more recent papers are cited, they could be considered in more detail. For instance, previous MR analyses have not only provided evidence for a causal association between retinal vasculometry and blood pressure (hence this approach has already been used), but also with CHD, which is not acknowledged here. Moreover, unlike earlier work, performance / validation of image processing modules against ground truths appears limited (with only 100 CFI), and no accompanying methods paper to validate the approach appears to be cited. In addition, 17 retinal vascular parameters are included but some are highly correlated, raising the possibility of epiphenomenon (i.e., that highly correlated parameters show similar patterns of association). There appears to be little a priori biological justification for the further measures, and their meaning, and whether these could give rise to spurious associations.

In summary, the authors may wish to consider inclusion of a replication data-set and give further detailed consideration of previous work (especially in UK Biobank). Optimisation of extracting pertinent and accurate retinal vascular characters is potentially key to homogenising findings from the increasing number of studies in this field.

This paper outlines an AI pipeline for retinal vasculometry (RV) extraction in a large, nationally representative, population-based study (UK Biobank). Color fundus photographs from 71,494 subjects with 130361 macular centred images were used to examine GWAS and phenotypic associations. While a lot of work has been done here, there are important weaknesses to be considered.

1. Critically there is no replication data-set, which is standard practice for this type of study.
2. In the absence of replication, findings should be more rigorously critiqued, particularly in comparison with papers which have been published recently using the same UK Biobank data-set. Note, these recently published studies (e.g., Zekevat SM et al, *Circulation*, 2022; Jian, *PLoS Genetics*, 2023) include replication in other data-sets, making earlier findings more robust.
3. As an example, heritability levels for retinal arteriole tortuosity (previously shown to be the strongest genetically determined RV trait) appear lower than those previously reported. This raises questions about the validity of the RV extraction process, and whether the optimal approach is being used. The process is important as it may impact on the degree to which findings from different studies can be compared. Moreover, validation of the method used appears limited, especially compared to other studies.
4. The added value of this study above recent work needs to be more clearly outlined.

Reviewer #2 (Remarks to the Author):

The study conducted by Bergmann et al. presents a comprehensive analysis of 17 different retinal morphological features using an automated pipeline. The research leverages data from the UK Biobank, including a large population and fundus images, to perform various analyses such as GWAS, genotype-phenotype correlation, pathway analysis, and Mendelian randomization. The study contributes to our understanding of the relationships between different retinal features at both the phenotypic and genotypic levels, as well as their associations with systemic diseases and traits. While the study is promising, there are several concerns that need to be addressed.

1. The manuscript lacks information regarding external validation of the automated pipeline for extracting the 17 retinal features and the quality control (QC) methods employed. Given the significant impact of image quality on certain retinal features, especially those susceptible to image artifacts, the stability and reliability of the model are crucial. The authors are suggested to provide details on any external datasets used for validation and demonstrate the robustness of the pipeline, particularly in handling variations in image quality.

2. The GWAS analysis results of retinal features presented in the manuscript need external validation to strengthen the study's credibility. Lack of external validation should be acknowledged as a study limitation.

3. More sensitivity analysis methods are suggested for the Mendelian randomization analysis to enhance the reliability of the results.

Reviewer #3 co-reviewed the article with Reviewer #1.

Reviewer #1a (Remarks to the Author):

This paper describes an automated system to extract retinal vessel indices from colour fundus images (CFI), with application to 130K CFI from 72K UK Biobank participants. The paper outlines GWAS and phenotype associations with retinal vessel traits, which show heritability of between 5 to 25%, potentially providing causal insights into disease outcomes and processes.

Unfortunately, the paper has a number of limitations. A major concern is the lack of an independent replication data-set to confirm the findings, which is increasingly becoming standard within this field.

We agree that it is common practice in Genome-wide Association Studies to seek for replication of the discovered associations using independent data. We had previously replicated some of our top associations for tortuosity using data from the *Swiss Kidney Project on Genes in Hypertension* (N = 397), and *OphthalmLaus* (N = 512) and demonstrated a significant correlation between the (directional) effect sizes from the UKBB and these much smaller studies (PMID: 37131961). Since the other traits exhibited lower heritability and less significant association signals than tortuosity, we anticipated that these cohorts would be too small to be sufficiently powered to attempt their replication. To address the reviewer's comment we needed a larger study, and we were very fortunate to be able to collaborate with the Rotterdam Study (RS), enabling the analysis of CFIs from N=8.1k genotyped subjects. Additionally, since *OphthalmLaus* in the meantime had increased its number of subjects (N=2.2k) we also validated our results on phenotypic correlations in this cohort.

We are happy to report that this effort allowed us to replicate our main findings. Our results are presented in Fig. 7 (reproduced below) of the main text (and with additional details in our Supplementary text "IDPs genetic replication"), specifically:

1. We reproduced the phenotypic correlation structure in both RS and *OphthalmLaus* participants (Fig. 7a). Analysis of RS data also resulted in highly significant RS-UKBB correlations between the estimates for trait heritabilities (Fig. 7b) and genetic correlations (Fig. 7c).
2. We replicated a large number of our genome-wide significant hits for the vast majority of traits. We show results for four IDPs in Fig. 7d and for all IDPs in the Suppl text "IDPs genetic replication".
3. We also found highly significant correlations between the corresponding effect sizes. We show results for four IDPs in Fig. 7e and for all IDPs in the Suppl text "IDPs genetic replication".
4. We replicated a large number of our genome-wide significant genes for the vast majority of traits. We show results for four IDPs in Fig. 7f and for all IDPs in the Suppl text "IDPs genetic replication".

These results clearly demonstrate that our phenotyping approach is robust and that our genotypic associations and subsequent analyses are not specific to the UKBB.

In the absence of a validation data-set, there should be greater emphasis on previous work, and this only receives cursory mention, which is largely assigned to supplemental material. There is a need for more direct comparison with previous work in the main text paper to provide validity, i.e., to what degree does this work mirror earlier work, particularly in UK Biobank? In my view supplemental material #11 could be built-upon and usefully included in the paper, with clearer consideration of the overlap (or lack of overlap) with loci identified previously.

With the replication study outlined in the previous response, we are now in a position to report which of the associations from UKBB data could be validated in the RS. Due to the sizable power, we could even identify novel genome-wide associations in the latter that replicate (but could not be discovered) in the UKBB. Nevertheless, we expanded the paragraph describing previously reported GWAS for retinal traits, rather than leaving this important information just in the Supplement:

Gene analysis of individual IDPs replicated some previously identified gene associations with explicit vascular phenotypes, namely tortuosity [Veluchamy et al. Arteriosclerosis (2019), Tomasoni et al. Ophthalmology Science (2023), Jiang et al. PLoS Genetics (2023)], fractal dimension, vascular density [Zekevat SM et al, Circulation (2022)], and vessel width [Jiang et al. PLoS Genetics (2023)]. [...] (see Suppl. Text “Comparison with previous GWAS”).

It is noteworthy that recent work has shown higher heritability for some retinal vascular traits (particularly for retinal arteriolar tortuosity) than reported here. Reasons for these differences could be considered further, and the degree to which this might reflect methodological approaches.

We have also been intrigued by this observation. We added the following paragraph on this matter to our Discussion:

Tortuosity and vascular density showed comparatively high heritability, consistent with some previous findings [Tomasoni et al. Ophthalmology Science (2023), Zekevat SM et al, Circulation (2022)]. A recent study [Jiang et al. PLoS Genetics (2023)] estimated heritability of retinal arteriolar tortuosity at 0.51 using UKBB data, substantially higher than our 0.25 estimate. Our point estimate in the RS is 0.22 ± 0.07 , which is consistent with our estimate from UKBB data. We note that here we used median tortuosity across all vessel segments independent of their caliber and length, even though tortuosity may vary as a function of blood vessel length and diameter, calling for more refined or stratified measures.

Indeed, considering only vessel segments in the fifth quintile for vessel length resulted in somewhat higher heritability (0.3). We plan an independent paper on stratified analyses (by segment length, diameter, or region) of IDPs to explore whether such refined IDPs can achieve higher heritability or predictive value for specific disease traits.

My view is that given recent papers published (especially in UK Biobank) the current work needs to clearly outline the added value of this work.

We now outline the major advances of our study with respect to previous work in our *Discussion*, in the following paragraph:

In this study, we established an automated analysis pipeline to extract 17 retinal vascular phenotypes from CFIs and applied it to over 130k CFIs of close to 72k UKBB subjects. While some of these phenotypes had previously been studied individually, our work is the first to provide a common reference. Our phenotyping procedure, automated and open access, enabled us to study jointly a large panel of retinal vascular phenotypes, some of which (temporal angles, central equivalents, number of bifurcations) were assessed for the first time in a large cohort. We provided a comparison of the phenotypic with genotypic correlation structures of these IDPs. We estimated their heritabilities, and elucidated associated genes and pathways, allowing us to identify common and disjoint genetic architectures. We studied associations of our IDPs with a spectrum of diseases and risk factors providing evidence of their complementarity for indicating specific disease risks. For validation, we reproduced the phenotypic correlation structure in two independent

cohorts, the RS and OphthalmoLaus, with 8.1k and 2.2k participants respectively, and validated numerous GWAS results in the RS. Importantly, the RS analysis pipeline was coded independently (to adapt to the specifics of their CFIs), such that these successful replication results provide strong evidence that our phenotyping is robust and not driven by cohort specific effects.

While these more recent papers are cited, they could be considered in more detail. For instance, previous MR analyses have not only provided evidence for a causal association between retinal vasculometry and blood pressure (hence this approach has already been used), but also with CHD, which is not acknowledged here.

We refer to previous MR analysis as follows:

Amongst the strongest potential reverse causal effects is the aforementioned decrease in BP due to a higher ratio of the central retinal equivalent. We found that individuals with higher arterial tortuosity tend to have higher BP, confirming previous findings [Jiang et al. PLoS Genetics, 2023].

It is true that we did not mention explicitly their MR results on CHD, as we did not consider this disease explicitly ourselves. We now included CHD in our MR analysis (see figure below) and refer to these results as follows:

The recent paper by Jiang et al. [Jiang et al. PLoS Genetics, 2023] also reported a causal effect of arterial tortuosity on Coronary Heart Disease (CHD) using the inverse variance weighted methodology. We were able to confirm this causal link of arterial tortuosity on CHD, while no other retinal trait was found to be causal (see Suppl. Text "Mendelian Randomization analysis between vascular IDPs and binary diseases").

In addition, 17 retinal vascular parameters are included but some are highly correlated, raising the possibility of epiphenomenon (i.e., that highly correlated parameters show similar patterns of association). There appears to be little a priori biological justification for the further measures, and their meaning, and whether these could give rise to spurious associations.

We agree that a high correlation between our phenotypic parameters can lead to epiphenomenon. For this reason "*We selected 17 representative IDPs from a broader set of 36 parameters to characterize each image [...] based on their associations with diseases and the reliability of measurement.*" (quoting our text in the introduction). We note that among our most highly correlated traits are the venous (VD) and arterial (AD) vascular densities ($r_{VD,AD} = 0.80$, after correction for covariates). VD and AD are clearly different phenotypes pertaining to physiologically distinct blood vessels, so their high correlation indicates that in many subjects arterial and venous vascularisation processes are strongly coupled. Interestingly, despite this high correlation, we found many more genes associated with VD (54) than AD (36) out of which only 26 are common. Another interesting observation is that while the number of bifurcations (NB) is strongly correlated to both vascular densities ($r_{NB,VD} = 0.88$, $r_{NB,AD} = 0.84$), it nevertheless is not redundant, since their associations with diseases sometimes differed. For instance, SBP and hypertension were associated with arterial vascular density and the number of bifurcations but not with venous vascular density.

In summary, the authors may wish to consider inclusion of a replication data-set and give further detailed consideration of previous work (especially in UK Biobank). Optimisation of extracting pertinent and accurate retinal vascular characters is potentially key to homogenising findings from the increasing number of studies in this field.

We thank the reviewer for the pertinent comments which we hope to have addressed adequately.

Reviewer #1b:

This paper outlines an AI pipeline for retinal vasculometry (RV) extraction in a large, nationally representative, population-based study (UK Biobank). Color fundus photographs from 71,494 subjects with 130361 macular centred images were used to examine GWAS and phenotypic associations. While a lot of work has been done here, there are important weaknesses to be considered.

1. Critically there is no replication data-set, which is standard practice for this type of study.

We added a replication analysis using data from the Rotterdam Study (RS). Please see our first answer to the previous reviewer for details.

2. In the absence of replication, findings should be more rigorously critiqued, particularly in comparison with papers which have been published recently using the same UK Biobank data-set. Note, these recently published studies (e.g., Zekevat SM et al, *Circulation*, 2022; Jian, *PLoS Genetics*, 2023) include replication in other data-sets, making earlier findings more robust.

Due to their relevance to our study, we now cite these works more prominently:

Gene analysis of individual IDPs replicated some previously identified gene associations with explicit vascular phenotypes, namely tortuosity [Veluchamy et al. Arteriosclerosis (2019), Tomasoni et al. Ophthalmology Science (2023), Jiang et al. PLoS Genetics (2023)], fractal dimension, vascular density [Zekevat SM et al, Circulation (2022)], and vessel width [Jiang et al. PLoS Genetics (2023)]. [...] (see Suppl. Text "Comparison with previous GWAS").

3. As an example, heritability levels for retinal arteriole tortuosity (previously shown to be the strongest genetically determined RV trait) appear lower than those previously reported. This raises questions about the validity of the RV extraction process, and whether the optimal approach is being used. The process is important as it may impact on the degree to which findings from different studies can be compared. Moreover, validation of the method used appears limited, especially compared to other studies.

The higher heritability estimate for arterial tortuosity was also mentioned by the previous reviewer (c.f. our response above). We note that our estimate from UKBB data (0.25) is consistent with that from RS data (0.22), while both are well below the estimate by Jian et al. (0.51). Importantly, the vessel segmentation models and the code for the RV extraction process implemented by the RS are different from the one we used for the UKBB fundus images, but nevertheless lead to consistent heritability estimates not only for tortuosity but also for other RV traits. As noted before restricting tortuosity measurements to vessel segments in the fifth quintile with respect to their length resulted in somewhat higher heritability (0.3), indicating that stringent vessel stratification may lead to more robust tortuosity measures. We hope that our open-access phenotyping platform will facilitate further research on this important aspect and encourage others to make their phenotyping pipeline freely available for researchers.

4. The added value of this study above recent work needs to be more clearly outlined.

We now outline the major advances of our study with respect to previous work in our *Discussion*, in the following paragraph:

In this study, we established an automated analysis pipeline to extract 17 retinal vascular phenotypes from CFIs and applied it to over 130k CFIs of close to 72k UKBB subjects. While some of these phenotypes had previously been studied individually, our work is the first to provide a common reference. Our phenotyping procedure, automated and open access, enabled us to study jointly a large panel of retinal vascular phenotypes, some of which (temporal angles, central equivalents, number of bifurcations) were assessed for the first time in a large cohort. We provided a comparison of the phenotypic with genotypic correlation structures of these IDPs. We estimated their heritabilities, and elucidated associated genes and pathways, allowing us to identify common and disjoint genetic architectures. We studied associations of our IDPs with a spectrum of diseases and risk factors providing evidence of their complementarity for indicating specific disease risks. For validation, we reproduced the phenotypic correlation structure in two independent cohorts, the RS and OphthalmoLaus, with 8.1k and 2.2k participants respectively, and validated numerous GWAS results in the RS. Importantly, the RS analysis pipeline was coded independently (to adapt to the specifics of their CFIs), such that these successful replication results provide strong evidence that our phenotyping is robust and not driven by cohort specific effects.

Reviewer #2 (Remarks to the Author):

The study conducted by Bergmann et al. presents a comprehensive analysis of 17 different retinal morphological features using an automated pipeline. The research leverages data from the UK Biobank, including a large population and fundus images, to perform various analyses such as GWAS, genotype-phenotype correlation, pathway analysis, and Mendelian randomization. The study contributes to our understanding of the relationships between different retinal features at both the phenotypic and genotypic levels, as well as their associations with systemic diseases and traits. While the study is promising, there are several concerns that need to be addressed.

1. The manuscript lacks information regarding external validation of the automated pipeline for extracting the 17 retinal features and the quality control (QC) methods employed. Given the significant impact of image quality on certain retinal features, especially those susceptible to image artifacts, the stability and reliability of the model are crucial. The authors are suggested to provide details on any external datasets used for validation and demonstrate the robustness of the pipeline, particularly in handling variations in image quality.

We thank the reviewer for commenting on this important aspect of our work, which had not been highlighted sufficiently, but is now detailed in the supplement:

For the UKBB data, we used the *Little-W-Net* for pixel-wise segmentation of blood vessels and annotating them as arteries or veins, and the *ARIA* tool for the annotation of vessel segments and extracting their midline and width measurements (supplement section 1.1). Both tools have been published and validated on CFIs similar to those available in the UKBB. Our pipeline for measuring tortuosity has also been published previously, including a comparison of different measures, highlighting the robustness of the distance factor, which was considered the measure of choice for the present study. Also, the measure for vascular densities (i.e. the fraction of pixels annotated as vessels) is well established, even though it depends of course on the quality of the primary segmentation (section 2.5). The non-trivial retinal features, for which we implemented new measurement methods are the number of bifurcations, the temporal angles, and the central retinal equivalents. For the two former, we validated our method using the DRIVE dataset and reported the details in our Supplementary Text "Validation of IDPs" (section 6). For the latter, we had to adapt the standard algorithm developed for CFIs with the optic disk at the center to those of the UKBB, which are fovea centered, i.e. have the optic disk at the side, and therefore often only allow to consider the temporal vasculature. While our method for this trait has not been formally validated, it is a straightforward adaptation of a well-established method. Importantly, it enabled automated measurements of central retinal equivalents and these data resulted in sizable heritability estimates, which we consider as indirect evidence for a robust and meaningful measurement. Of note our GWAS for the central equivalents led to genome-wide significant associations, some of which were replicated in the RS data using an independently developed phenotyping pipeline.

As for the image QC we relied on a published method [Zekevat SM et al, *Circulation*, 2022]. Additionally, we tried using different QC cutoffs (figure below and in Supplementary Text "Quality control threshold effect on results" in section 18) to see

if this cutoff could have a big impact on our heritability estimates:

2. The GWAS analysis results of retinal features presented in the manuscript need external validation to strengthen the study's credibility. Lack of external validation should be acknowledged as a study limitation.

We added a replication analysis. Please see our first answer to the previous reviewers for details.

3. More sensitivity analysis methods are suggested for the Mendelian randomization analysis to enhance the reliability of the results.

We extended our MR analysis, which now includes

- an assessment of instrument strength using F statistics, confirming that we have suitable instruments for all trait pairs
- implementation of additional methods, namely MR-Egger, weighted median, and weighted mode MR, indicated that estimated causal effects were consistent in the direction across the four methods whenever significant
- estimation of MR-Egger intercepts (to be non-significant), suggesting the absence of sizable pleiotropy
- leave-one-out analyses, showing that our estimates were not biased by any single instrument

Detailed explanations are provided in our revised Methods section.

REVIEWERS' COMMENTS

Reviewer #3 (Remarks to the Author):

This paper is much improved with the inclusion of the replication data-sets, which were lacking in the previous version. I have a few additional comments...

Introduction - There are other examples of open source retinal vascular phenotyping (e.g., <https://doi.org/10.1167/tvst.11.7.12>). Also examples of AI retinal feature detection approaches to predict vascular outcomes using UK Biobank (e.g., <https://doi.org/10.1136/bjo-2022-321842>). Reference 48 measures both retinal vessel diameters and tortuosity, importantly for arterioles and venules separately, which other studies do not (e.g., reference 49).

The camera used in UK Biobank was the Topcon 3D-OCT 1000 Mark 2, not the Triton.

Note, the authors did not respond to my previous comment 'Moreover, unlike earlier work, performance / validation of image processing modules against ground truths appears limited (with only 100 CFI), and no accompanying methods paper to validate the approach appears to be cited'. This is important as it could partially explain why the heritability of arteriolar tortuosity in this study is so much lower than that previously found (i.e. 0.25 vs 0.51 found in reference 48). In my view, this raises concerns over the measure and the need for further work; this could be included as a limitation.

Baseline characteristics of those analyzed from UK Biobank given in Supplemental Material (i.e., ~68,500 of ~84,800), but no demographic data on those not processed to assess selection (i.e., on the ~16,300).

Reviewer #4 (Remarks to the Author):

The authors have carefully addressed all comments and concerns presented in the initial evaluation of the manuscript. I am in particular pleased with the additional data presented for external validation.

Reviewer #4 (Remarks on code availability):

Not able to access.

Response REVIEWERS' COMMENTS:

Reviewer #3 (Remarks to the Author):

This paper is much improved with the inclusion of the replication data-sets, which were lacking in the previous version. I have a few additional comments...

Introduction - There are other examples of open source retinal vascular phenotyping (e.g., <https://doi.org/10.1167/tvst.11.7.12>). Also examples of AI retinal feature detection approaches to predict vascular outcomes using UK Biobank (e.g., <https://doi.org/10.1136/bjo-2022-321842>).

We thank the reviewer for the feedback. We acknowledge the relevance of the two papers suggested and we now cite them in our Introduction:

"Furthermore, the software used for vascular phenotyping is usually not openly accessible, with two recent exceptions~\cite{zhou2022automorph, fhima2023pvbm}... DL approaches have been used to directly predict health-relevant phenotypes from retinal images~\cite{poplins2018prediction, li2021applications, rudnicka2022artificial}."

Reference 48 measures both retinal vessel diameters and tortuosity, importantly for arterioles and venules separately, which other studies do not (e.g., reference 49).

Yes, Jiang et al. (and also Tomassoni et al. and Veluchamy et al.) measured retinal vessel parameters for both arterioles and venules separately, while others did not. We now acknowledge this by citing the former after our sentence:

"First, the image is processed at the level of pixels to identify which of them represents blood vessels (possibly distinguishing between arteries and veins~\cite{jiang2023gwas, tomasoni2023genome, veluchamy2019novel}."

The camera used in UK Biobank was the Topcon 3D-OCT 1000 Mark 2, not the Triton.

We thank the reviewer for noticing this and we have changed this from the main text.

"Standard retinal 45° CFIs were captured using a Topcon 3D-OCT 1000 Mark II."

Note, the authors did not respond to my previous comment 'Moreover, unlike earlier work, performance / validation of image processing modules against ground truths appears limited (with only 100 CFI), and no accompanying methods paper to validate the approach appears to be cited'. This is important as it could partially explain why the heritability of arteriolar tortuosity in this study is so much lower than that previously found (i.e. 0.25 vs 0.51 found in reference 48). In my view, this raises concerns over the measure and the need for further work; this could be included as a limitation.

We apologise for not addressing the validation of the image processing module sufficiently. We now revised our Methods section to state explicitly that the Little W-Net, which we used for the segmentation of CFIs from the UKBB, had already been validated by the authors:

"Raw CFIs were segmented into pixel-wise segmentations of arteries and veins using the DL model LUNET~\cite{galdran2022state}, which had been trained **and validated** on the publicly available DRIVE dataset."

The 100 CFIs that we mentioned are a small subset that we only used to check that the (already validated) Little W-Net also works well on CFIs from the UKBB. In any case our main argument for the robustness of our analysis now lies in the replication of our UKBB results in the Rotterdam Study, and this includes, as we noted, a consistent heritability estimate for tortuosity.

Baseline characteristics of those analyzed from UK Biobank given in Supplemental Material (i.e., ~68,500 of ~84,800), but no demographic data on those not processed to assess selection (i.e., on the ~16,300).

We provided Disease incidence rates for different QC stringencies in Suppl. Figure 33: The green bars for the 20th percentile removed corresponds to the QC we applied. The figure show that the incident rates for all disease decrease with increasing QC stringency, yet for a cutoff removing the 20% of the images with the lowest quality this decrease is not very dramatic.

Reviewer #4 (Remarks to the Author):

The authors have carefully addressed all comments and concerns presented in the initial evaluation of the manuscript. I am in particular pleased with the additional data presented for external validation.

We thank the reviewer for the positive feedback.

Reviewer #4 (Remarks on code availability):

Not able to access.

Our code is now available in the link of the publication: <https://github.com/BergmannLab/retina-phenotypes>

One of the reviewers got in touch after the decision was sent and provided the following comments: "The authors responded well to all comments raised but for R2.12 must acknowledge that all their subgroups (age, sex and ethnicity) had small numbers with serious disease (not just for other smaller subgroups, not sure which these would be) and is an important limitation of this study."

We thank the reviewer for the feedback. In this study we did not split by subgroups. We used age, sex, ethnicity (with the genetic principal components), as well as other parameters as covariates, which were subtracted from the phenotypes (as explained in Figure 1a, and in the methods section). The number of subjects with serious diseases can be found in the y-axis of Figure 4, and indeed the number of cases varied highly among diseases, the smallest being 500 for subjects who had pulmonary embolism (PE). This variability of the number of cases across the different diseases can influence the results, and for that, we now include this as a limitation:

"the variability in the number of cases with severe diseases, as shown in Figure~\ref{fig_phenot_diseases_cox_MLR}, may influence the results."